# Flow map matching with stochastic interpolants: A mathematical framework for consistency models

**Nicholas M. Boffi**\*                                                     *boffi@cims.nyu.edu*
*Courant Institute of Mathematical Sciences*

**Michael S. Albergo**\*                                                    *albergo@nyu.edu*
*Courant Institute of Mathematical Sciences*

**Eric Vanden-Eijnden**                                                     *eve2@cims.nyu.edu*
*Courant Institute of Mathematical Sciences*

**Reviewed on OpenReview:** *https://openreview.net/forum?id=cqDH0e6ak2*

## Abstract

Generative models based on dynamical equations such as flows and diffusions offer exceptional sample quality, but require computationally expensive numerical integration during inference. The advent of consistency models has enabled efficient one-step or few-step generation, yet despite their practical success, a systematic understanding of their design has been hindered by the lack of a comprehensive theoretical framework. Here we introduce Flow Map Matching (FMM), a principled framework for learning the two-time flow map of an underlying dynamical generative model, thereby providing this missing mathematical foundation. Leveraging stochastic interpolants, we propose training objectives both for distillation from a pre-trained velocity field and for direct training of a flow map over an interpolant or a forward diffusion process. Theoretically, we show that FMM unifies and extends a broad class of existing approaches for fast sampling, including consistency models, consistency trajectory models, and progressive distillation. Experiments on CIFAR-10 and ImageNet-32 highlight that our approach can achieve sample quality comparable to flow matching while reducing generation time by a factor of 10-20.

## 1 Introduction

In recent years, diffusion models (Song et al., 2020; Ho et al., 2020; Sohl-Dickstein et al., 2015; Song and Ermon, 2020a;b) have achieved state of the art performance across diverse modalities, including image (Dhariwal and Nichol, 2021; Rombach et al., 2022; Esser et al., 2024), audio (Popov et al., 2021; Jeong et al., 2021; Huang et al., 2022; Lu et al., 2022a), and video (Ho et al., 2022a;b; Blattmann et al., 2023; Wu et al., 2023). These models belong to a broader class of approaches including flow matching (Lipman et al., 2022), rectified flow (Liu et al., 2022a), and stochastic interpolants (Albergo and Vanden-Eijnden, 2022; Albergo et al., 2023a), which construct a path in the space of measures between a base and a target distribution by specifying an explicit mapping between samples from each. This construction yields a dynamical transport equation governing the evolution of the time-dependent probability measure along the path. The generative modeling problem then reduces to learning a velocity field that accomplishes this transport, which provides an efficient and stable training paradigm.

At sample generation time, models in this class generate data by smoothly transforming samples from the base into samples from the target through numerical integration of a differential equation. While effective, the number of integration steps required to produce high-quality samples incurs a cost that can limit real-time applications (Chi et al., 2024). Comparatively, one-step models such as generative adversarial

---

\*Authors contributed equally; author ordering determined randomly.

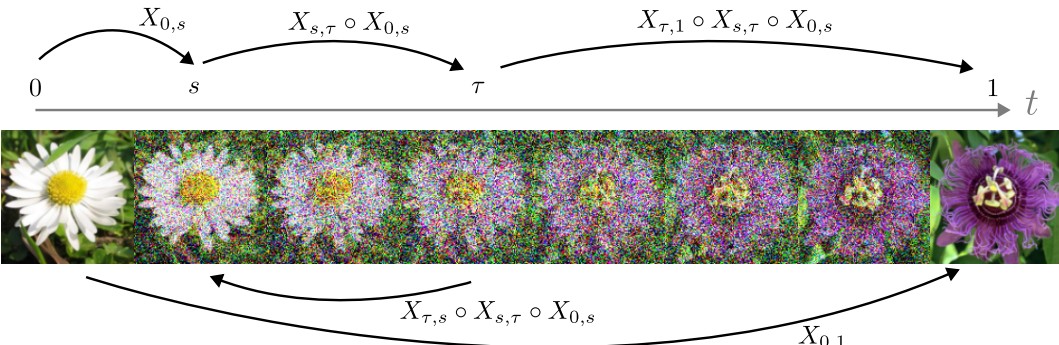

**Figure 1: Overview of Flow Map Matching.** Our approach learns the two-time flow map $X_{s,t}$ that transports the solution of an ordinary differential equation from time $s$ to time $t$. Unlike methods that learn instantaneous velocity fields, this bidirectional map can be used to build an integrator with arbitrary discretization. The integrator is exact in theory, and –crucially– its number of steps can be adjusted post-training to balance accuracy and computational efficiency. The flow map can be distilled from a known velocity field or learned directly, and supports arbitrary base distributions, as illustrated here with image-to-image translation.

networks (Goodfellow et al., 2014; 2020; Creswell et al., 2018) are notoriously difficult to train (Metz et al., 2017; Arjovsky et al., 2017), but can be orders of magnitude more efficient to sample, because they only require a single network evaluation. As a result, there has been significant recent research effort focused on maintaining the stable training of diffusions while reducing the computational burden of inference (Karras et al., 2022). Towards this goal, we introduce Flow Map Matching—a theoretical framework for learning the two-time flow map of a probability flow. Specifically, our primary contributions can be summarized as:

- **A mathematical framework.** We characterize the flow map theoretically by identifying its fundamental mathematical properties, which we show immediately lead to distillation-based algorithms for estimating the flow map from a pre-trained velocity field. Using our characterization, we establish connections between our framework and recent approaches for learning consistency models (Song et al., 2023; Song and Dhariwal, 2023; Kim et al., 2024a) and for progressive distillation of a diffusion model (Salimans and Ho, 2022; Zheng et al., 2023).

- **Lagrangian vs Eulerian learning.** We develop a novel *Lagrangian* objective for distilling a flow map from a pre-trained velocity field, which performs well across our experiments. This loss is complementary to a related *Eulerian* loss, which we prove is the continuous-time limit of consistency distillation (Song et al., 2023). We show that both the Lagrangian and Eulerian losses control the Wasserstein distance between the teacher and the student models.

- **Direct training.** We extend our Lagrangian loss to design a novel direct training objective for flow maps that eliminates the need for a pre-trained velocity field. We also discuss how to modify the Eulerian loss to allow for direct training of the map, and identify some caveats with this procedure. Our approach elucidates how the standard training signal for flow matching can be adapted to capture the more global signal needed to train the flow map.

- **Numerical experiments.** We validate our theoretical framework through experiments on CIFAR-10 and ImageNet $32 \times 32$, where we achieve high sample quality at significantly reduced computational cost. In particular, we highlight that learning the two-time flow map enables post-training adjustment of sampling steps, allowing practitioners to systematically trade accuracy for computational efficiency.

## 2 Related Work

**Dynamical transport of measure.** Our approach is built upon the modern perspective of generative modeling based on dynamical transport of measure. Grounded in the theory of optimal transport (Villani,

2009; Benamou and Brenier, 2000; Santambrogio, 2015), these models originate at least with (Tabak and Vanden-Eijnden, 2010; Tabak and Turner, 2013), but have been further developed by the machine learning community in recent years (Rezende and Mohamed, 2015; Dinh et al., 2017; Grathwohl et al., 2018; Chen et al., 2019). A breakthrough in this area originated with the appearance of score-based diffusion models (Song et al., 2020; Song and Ermon, 2020a;b), along with related denoising diffusion probabilistic models (Ho et al., 2020; Sohl-Dickstein et al., 2015). These methods generate samples by learning to time-reverse a stochastic differential equation with stationary density given by a Gaussian. More recent approaches such as flow matching (Lipman et al., 2022), rectified flow (Liu et al., 2022a;b), and stochastic interpolants (Albergo and Vanden-Eijnden, 2022; Albergo et al., 2023a; Ma et al., 2024; Chen et al., 2024) similarly construct connections between the base density and the target, but allow for arbitrary base densities and provide a greater degree of flexibility in the construction of the connection.

**Reducing simulation costs.** There has been significant recent interest in reducing the cost associated with solving an ODE or an SDE for a generative model based on dynamical measure transport. One such approach, pioneered by rectification (Liu et al., 2022a;b), is to try to straighten the paths of the probability flow, so as to enable more efficient adaptive integration. In the limit of optimal transport, the paths become straight lines and the integration can be performed in a single step. A second approach is to introduce *couplings* between the base and the target, such as by computing the optimal transport over a minibatch (Pooladian et al., 2023; Tong et al., 2023), or by using data-dependent couplings (Albergo et al., 2023b), which can simplify both training and sampling. A third approach has been to design hand-crafted numerical solvers tailored for diffusion models (Karras et al., 2022; Zhang and Chen, 2023; Jolicoeur-Martineau et al., 2021; Liu et al., 2022c; Lu et al., 2022b), or to learn these solvers directly (Watson et al., 2021; 2022; Nichol and Dhariwal, 2021) to maximize efficiency. *Instead, we propose to learn the flow map directly, which avoids estimating optimal transport maps and can overcome the inherent limitations of numerical integration.*

**Consistency models.** Most related to our approach are the classes of one-step models based on *distillation* or *consistency*; we give an explicit mapping between these techniques and our own in Appendix C. Consistency models (Song et al., 2023) have been introduced as a new class of generative models that can either be distilled from a pre-trained diffusion model or trained directly, and are related to several notions of *consistency* of the score model that have appeared in the literature (Lai et al., 2023a;b; Shen et al., 2022; Boffi and Vanden-Eijnden, 2023; Daras et al., 2023). These models learn a one-step map from noise to data, and can be seen as learning a single-time flow map. While they can perform very well, consistency models do not benefit from multistep sampling, and exhibit training difficulties that mandate delicate hyperparameter tuning (Song and Dhariwal, 2023). By contrast, we learn a two-time flow map, which enables us to smoothly benefit from multistep sampling.

**Extensions of consistency models.** Consistency trajectory models (Kim et al., 2024a) were later introduced to improve multistep sampling and to enable the student to surpass the performance of the teacher. Similar to our approach, these models learn a two-time flow map, but do so using a very different loss that incorporates challenging adversarial training. Generalized consistency trajectory models (Kim et al., 2024b) extend this approach to the stochastic interpolant setting, but use the consistency trajectory loss, and do not introduce the Lagrangian perspective considered here. Bidirectional consistency models (Li and He, 2024) learn a two-time invertible flow map similar to our method, but do so in the score-based diffusion setting, and do not leverage our Lagrangian approach; all of these existing consistency models can be seen as a special case of our formalism.

**Progressive and operator distillation.** Neural operator approaches (Zheng et al., 2023) learn a one-time flow map from noise to data, but do so by first generating a dataset of trajectories from the probability flow. Progressive distillation (Salimans and Ho, 2022) and knowledge distillation (Luhman and Luhman, 2021) techniques aim to convert a diffusion model into an equivalent model with fewer samples by matching several steps of the original diffusion model. These approaches are related to our flow map distillation scheme, though the object we distill is fundamentally different.

# 3 Flow Map Matching

The central object in our method is the *flow map*, which maps points along trajectories of solutions to an ordinary differential equation (ODE). Our focus in this work is primarily on probability flow ODEs that arise in generative models, such as those constructed using flow matching via stochastic interpolants or score-based diffusion models (see Appendix A for a concise, self-contained review of these approaches). While many of our results apply more generally to other ODEs, we present our definitions and theoretical results in this specific context to highlight their relevance to generative modeling. All proofs of the statements made in this section are provided in Appendix B, with some additional theoretical results and connections to existing consistency and distillation techniques given in Appendix C.

## 3.1 Stochastic interpolants and probability flows

We first recall the main components of the stochastic interpolant framework (Albergo and Vanden-Eijnden, 2022; Albergo et al., 2023a), which we use to construct the probability flow ODEs central to our study. We begin by giving the definition of a stochastic interpolant.

**Definition 3.1** (Stochastic Interpolant). *The stochastic interpolant $I_t$ between probability densities $\rho_0$ and $\rho_1$ is the stochastic process given by*

$$I_t = \alpha_t x_0 + \beta_t x_1 + \gamma_t z, \tag{3.1}$$

*where $\alpha, \beta, \gamma^2 \in C^1([0,1])$ satisfy $\alpha_0 = \beta_1 = 1$, $\alpha_1 = \beta_0 = 0$, and $\gamma_0 = \gamma_1 = 0$. In (3.1), $(x_0, x_1)$ is drawn from a coupling $(x_0, x_1) \sim \rho(x_0, x_1)$ that satisfies the marginal constraints $\int_{\mathbb{R}^d} \rho(x_0, x_1) dx_0 = \rho_1(x_1)$ and $\int_{\mathbb{R}^d} \rho(x_0, x_1) dx_1 = \rho_0(x_0)$. Moreover, $z \sim \mathsf{N}(0, Id)$ with $z \perp (x_0, x_1)$.*

Theorem 3.6 of Albergo et al. (2023a) shows that the stochastic interpolant given in Definition 3.1 specifies an underlying probability flow ODE, as we now recall.

**Proposition 3.2** (Probability Flow). *For all $t \in [0,1]$, the probability density of $I_t$ is the same as the probability density of the solution to*

$$\dot{x}_t = b_t(x_t), \qquad x_{t=0} = x_0 \sim \rho_0, \tag{3.2}$$

*where $b : [0,1] \times \mathbb{R}^d \to \mathbb{R}^d$ is the time-dependent velocity field (or drift) given by*

$$b_t(x) = \mathbb{E}[\dot{I}_t | I_t = x]. \tag{3.3}$$

*In (3.3), $\mathbb{E}[\cdot | I_t = x]$ denotes an expectation over the coupling $(x_0, x_1) \sim \rho(x_0, x_1)$ and $z \sim \mathsf{N}(0, Id)$ conditioned on the event $I_t = x$.*

Although (3.2) is deterministic for any single trajectory, taken together, its solutions are random because the initial conditions are sampled from $\rho_0$. We denote by $\rho_t = \text{Law}(x_t)$ the density of these solutions at time $t$. The drift $b$ can be learned efficiently in practice by solving a square loss regression problem (Albergo et al., 2023a)

$$b = \operatorname*{argmin}_{\hat{b}} \int_0^1 \mathbb{E}\big[|\hat{b}_t(I_t) - \dot{I}_t|^2\big] dt, \tag{3.4}$$

where $\mathbb{E}$ denotes an expectation over the coupling $(x_0, x_1) \sim \rho(x_0, x_1)$ and $z \sim \mathsf{N}(0, Id)$.

A canonical choice when $\rho_0 = \mathsf{N}(0, Id)$ considered in Albergo and Vanden-Eijnden (2022) corresponds to $\alpha_t = 1 - t$, $\beta_t = t$, and $\gamma_t = 0$, which recovers flow matching (Lipman et al., 2022) and rectified flow (Liu et al., 2022a). The choice $\alpha_t = 0$, $\beta_t = t$ and $\gamma_t = \sqrt{1 - t^2}$ corresponds to a variance-preserving diffusion model with the time-rescaling $t = -\log \tau$ where $\tau \in [0, \infty)$ is the usual diffusion time[1]. A variance-exploding diffusion model may be obtained by taking $\alpha_t = 0$, $\beta_t = 1$, and $\gamma_t = T - t$ with $t \in [0,1]$ and where $\tau = T - t$ is the usual diffusion time, though this violates the boundary conditions in Definition 3.1. For more details about the connection between stochastic interpolants and diffusion models, we again refer the reader to Appendix A.

---

[1]Note that $\gamma_0 = 1$ in this case, so that $I_0 = z$

### 3.2 Flow map: definition and characterizations

To generate a sample from the target density $\rho_1$, we can draw an initial point from $\rho_0$ and numerically integrate the probability flow ODE (3.2) over the interval $t \in [0,1]$. While this approach produces high-quality samples, it typically requires numerous integration steps, making inference computationally expensive—particularly when $b_t$ is parameterized by a complex neural network. Here, we bypass this numerical integration by estimating the two-time flow map, which lets us take jumps of arbitrary size. To do so, we require the following regularity assumption, which ensures (3.2) has solutions that exist and are unique (Hartman, 2002).

**Assumption 3.3.** *The drift satisfies the one-sided Lipschitz condition*

$$\exists \ C_t \in L^1[0,1] \ : \quad (b_t(x) - b_t(y)) \cdot (x - y) \leqslant C_t |x - y|^2 \quad \textit{for all } (t, x, y) \in [0,1] \times \mathbb{R}^d \times \mathbb{R}^d. \tag{3.5}$$

With Assumption 3.3 in hand, we may now define the central object of our study.

**Definition 3.4** (Flow Map). *The flow map $X_{s,t} : \mathbb{R}^d \to \mathbb{R}^d$ for (3.2) is the unique map such that*

$$X_{s,t}(x_s) = x_t \ \textit{for all } (s,t) \in [0,1]^2, \tag{3.6}$$

*where $(x_t)_{t \in [0,1]}$ is any solution to the ODE (3.2).*

The flow map in Definition 3.4 can be seen as an integrator for (3.2) where the step size $t - s$ may be chosen arbitrarily. In addition to the defining condition (3.6), we now highlight some of its useful properties.

**Proposition 3.5.** *The flow map $X_{s,t}(x)$ is the unique solution to the Lagrangian equation*

$$\partial_t X_{s,t}(x) = b_t(X_{s,t}(x)), \qquad X_{s,s}(x) = x, \tag{3.7}$$

*for all $(s, t, x) \in [0,1]^2 \times \mathbb{R}^d$. In addition, it satisfies*

$$X_{t,\tau}(X_{s,t}(x)) = X_{s,\tau}(x) \tag{3.8}$$

*for all $(s, t, \tau, x) \in [0,1]^3 \times \mathbb{R}^d$. In particular $X_{s,t}(X_{t,s}(x)) = x$ for all $(s, t, x) \in [0,1]^2 \times \mathbb{R}^d$, i.e. the flow map is invertible.*

Proposition 3.5 shows that, given an $x_0 \sim \rho_0$, we can use the flow map to generate samples from $\rho_t$ for any $t \in [0,1]$ in one step via $x_t = X_{0,t}(x_0) \sim \rho_t$.

The composition relation (3.8) is the *consistency property*, here stated over two times, that gives consistency models their name (Song et al., 2023). This relation shows that we can also generate samples in multiple steps using $x_{t_k} = X_{t_{k-1},t_k}(x_{t_{k-1}}) \sim \rho_{t_k}$ for any set of discretization points $(t_0, \ldots, t_K)$ with $t_k \in [0,1]$ and $K \in \mathbb{N}$. We refer to (3.7) as the *Lagrangian equation* because it is defined in a frame of reference that moves with $X_{s,t}(x)$.

The flow map $X_{s,t}$ also obeys an alternative *Eulerian equation* that is defined at any fixed point $x \in \mathbb{R}^d$ and which involves a derivative with respect to $s$. To derive this equation, note that since $X_{s,t}(x_s) = x_t$ by (3.6) we have $(d/ds)X_{s,t}(x_s) = 0$, which by the chain rule can be written as

$$\frac{d}{ds}X_{s,t}(x_s) = \partial_s X_{s,t}(x_s) + b_s(x_s) \cdot \nabla X_{s,t}(x_s) = 0 \tag{3.9}$$

where we used (3.2) to set $\dot{x}_s = b_s(x_s)$. Evaluating this equation at $x_s = x$ gives the announced result.

**Proposition 3.6.** *The flow map $X_{s,t}$ is the unique solution of the Eulerian equation,*

$$\partial_s X_{s,t}(x) + b_s(x) \cdot \nabla X_{s,t}(x) = 0, \qquad X_{t,t}(x) = x, \tag{3.10}$$

*for all $(s, t, x) \in [0,1]^2 \times \mathbb{R}^d$.*

### 3.3 Distillation of a known velocity field

The Lagrangian equation (3.7) in Proposition 3.5 leads to a distillation loss that can be used to learn a flow map for a probability flow with known right-hand side $b$, as we now show.

**Corollary 3.7** (Lagrangian map distillation). *Let $w_{s,t} \in L^1([0,1]^2)$ be a weight function satisfying $w_{s,t} > 0$ and let $I_s$ be the stochastic interpolant defined in (3.1). Then the flow map is the global minimizer over $\hat{X}$ of the loss*

$$\mathcal{L}_{\mathsf{LMD}}(\hat{X}) = \int_{[0,1]^2} w_{s,t} \mathbb{E}\big[|\partial_t \hat{X}_{s,t}(I_s) - b_t(\hat{X}_{s,t}(I_s)|^2\big] ds dt, \qquad (3.11)$$

*subject to the boundary condition that $\hat{X}_{s,s}(x) = x$ for all $x \in \mathbb{R}^d$ and $s \in [0,1]$. In (3.11), $\mathbb{E}$ denotes an expectation over the coupling $(x_0, x_1) \sim \rho(x_0, x_1)$ and $z \sim \mathsf{N}(0, Id)$.*

The time integrals in (3.11) can also be written as an expectation over $(s,t) \sim w_{s,t}$ (properly normalized):

$$\mathcal{L}_{\mathsf{LMD}}(\hat{X}) = \mathbb{E}_{(s,t) \sim w_{s,t}} \mathbb{E}\big[|\partial_t \hat{X}_{s,t}(I_s) - b_t(\hat{X}_{s,t}(I_s))|^2\big]. \qquad (3.12)$$

We also note that the result of Corollary 3.7 remains true if we replace $I_s$ by any process $\hat{x}_s$ with density $\hat{\rho}_s \neq \rho_s$, as long as it is strictly positive everywhere. In practice, it is convenient to use $I_s$ as it guarantees that we learn the flow map where we typically need to evaluate it. Moreover, using $I_s$ avoids the need to generate a dataset from a pre-trained model, so that the objective (3.11) can be evaluated empirically in a simulation-free fashion.

When applied to a pre-trained flow model, Corollary 3.7 can be used to train a new, few-step generative model with performance that matches the performance of the teacher. When $\hat{X}_{s,t}$ is parameterized by a neural network, the partial derivative with respect to $t$ can be computed efficiently at the same time as the computation of $\hat{X}_{s,t}$ using forward-mode automatic differentiation. This procedure is summarized in Algorithm 1.

For simplicity, Corollary 3.7 is stated for $w_{s,t} > 0$ (e.g. $w_{s,t} = 1$), so that we can estimate the map $X_{s,t}$ and its inverse $X_{t,s}$ for all $(s,t) \in [0,1]$. Nevertheless, this weight can also be adjusted to learn the map for different pairs $(s,t)$ of interest. For example, if we only want to estimate the forward map with $s \leqslant t$, then we can set $w_{s,t} = 1$ if $s \leqslant t$ and $w_{s,t} = 0$ otherwise.

By squaring the left hand-side of the Eulerian equation (3.10) in Proposition 3.6, we may construct a second loss function for distillation.[2]

**Corollary 3.8** (Eulerian map distillation). *Let $w_{s,t}$ and $I_s$ be as in Corollary 3.7. Then the flow map is the global minimizer over $\hat{X}$ of the loss*

$$\mathcal{L}_{\mathsf{EMD}}(\hat{X}) = \int_{[0,1]^2} w_{s,t} \mathbb{E}\big[|\partial_s \hat{X}_{s,t}(I_s) + b_s(I_s) \cdot \nabla \hat{X}_{s,t}(I_s)|^2\big] ds dt, \qquad (3.13)$$

*subject to the boundary condition $\hat{X}_{s,s}(x) = x$ for all $x \in \mathbb{R}^d$ and for all $s \in \mathbb{R}$.*

As in the discussion after Corollary 3.7, here we may also replace $I_s$ by any process $\hat{x}_s$ with a strictly positive density. Writing the time integrals in (3.13) as an expectation over $(s,t) \sim w_{s,t}$ gives

$$\mathcal{L}_{\mathsf{LMD}}(\hat{X}) = \mathbb{E}_{(s,t) \sim w_{s,t}} \mathbb{E}\big[\big|\partial_s \hat{X}_{s,t}(I_s) + b_s(I_s) \cdot \nabla \hat{X}_{s,t}(I_s)\big|^2\big]. \qquad (3.14)$$

We summarize a training procedure based on Corollary 3.8 in Algorithm 2.

In Appendix C, we demonstrate how the preceding results connect with existing distillation-based approaches. In particular, when $b_t(x)$ is the velocity of the probability flow ODE associated with a diffusion model, Corollary 3.8 recovers the continuous-time limit of consistency distillation (Song et al., 2023; Song and Dhariwal, 2023) and consistency trajectory models (Kim et al., 2024a), while Corollary 3.7 is new.

---

[2]In (3.10), the term $(b_s(x) \cdot \nabla X_{s,t}(x))_i = \sum_{j=1}^d [b_s(x)]_j \partial_{x_j} [X_{s,t}(x)]_i = [\nabla X_{s,t}(x) \cdot b_s(x)]_i$ corresponds to a Jacobian-vector product that can be computed efficiently using forward-mode automatic differentiation.

---

**Algorithm 1:** Lagrangian map distillation (LMD)

**input:** Interpolant coefficients $\alpha_t, \beta_t, \gamma_t$; pre-trained velocity $b_t$, weight function $w_{s,t}$, batch size $M$.

**repeat**

> Draw batch $(s_i, t_i, x_0^i, x_1^i, z_i)_{i=1}^M$ from $w_{s,t} \times \rho(x_0, x_1) \times \mathsf{N}(z; 0, Id)$.
> Compute $I_{t_i} = \alpha_{t_i} x_0^i + \beta_{t_i} x_1^i + \gamma_{t_i} z_i$ and $\dot{I}_{t_i} = \dot{\alpha}_{t_i} x_0^i + \dot{\beta}_{t_i} x_1^i + \dot{\gamma}_{t_i} z_i$.
> Compute $\hat{\mathcal{L}}_{\mathsf{LMD}} = \frac{1}{M} \sum_{i=1}^M |\partial_t \hat{X}_{s_i, t_i}(I_{s_i}) - b_{t_i}(\hat{X}_{s_i, t_i}(I_{s_i}))|^2$.
> Take gradient step on $\hat{\mathcal{L}}_{\mathsf{LMD}}$ to update $\hat{X}$.

**until** *converged*;

**output:** Flow map $\hat{X}$.

---

**Algorithm 2:** Eulerian map distillation (EMD)

**input:** Interpolant coefficients $\alpha_t, \beta_t, \gamma_t$; pre-trained velocity $b_t$, weight function $w_{s,t}$, batch size $M$.

**repeat**

> Draw batch $(s_i, t_i, x_0^i, x_1^i, z_i)_{i=1}^M$ from $w_{s,t} \times \rho(x_0, x_1) \times \mathsf{N}(z; 0, Id)$.
> Compute $I_{t_i} = \alpha_{t_i} x_0^i + \beta_{t_i} x_1^i + \gamma_{t_i} z_i$ and $\dot{I}_{t_i} = \dot{\alpha}_{t_i} x_0^i + \dot{\beta}_{t_i} x_1^i + \dot{\gamma}_{t_i} z_i$.
> Compute $\hat{\mathcal{L}}_{\mathsf{EMD}} = \frac{1}{M} \sum_{i=1}^M |\partial_s \hat{X}_{s_i, t_i}(I_{s_i}) + \nabla \hat{X}_{s_i, t_i}(I_{s_i}) b_{s_i}(I_{s_i})|^2$.
> Take gradient step on $\hat{\mathcal{L}}_{\mathsf{EMD}}$ to update $\hat{X}$.

**until** *converged*;

**output:** Flow map $\hat{X}$.

---

### 3.4 Wasserstein control

In this section, we show that the Lagrangian and Eulerian distillation losses (3.11) and (3.13) control the Wasserstein distance between the density $\rho_t$ of the teacher flow model and the density $\hat{\rho}_t = \hat{X}_{0,t} \sharp \rho_0$ of the pushforward of $\rho_0$ under the learned flow map (that is, $\hat{\rho}_t$ is the density of $\hat{X}_{0,t}(x_0)$ with $x_0 \sim \rho_0$). When combined with the Wasserstein bound in Albergo and Vanden-Eijnden (2022), the following results also imply a bound on the Wasserstein distance between the data density and the pushforward density for the learned flow map in the case where $b$ is a pre-trained stochastic interpolant or diffusion model. We begin by stating our result for Lagrangian distillation.

**Proposition 3.9** (Lagrangian error bound). *Let $X_{s,t} : \mathbb{R}^d \to \mathbb{R}^d$ denote the flow map for $b$, and let $\hat{X}_{s,t} : \mathbb{R}^d \to \mathbb{R}^d$ denote an approximate flow map. Given $x_0 \sim \rho_0$, let $\hat{\rho}_1$ be the density of $\hat{X}_{0,1}(x_0)$ and let $\rho_1$ be the target density of $X_{0,1}(x_0)$. Then,*

$$W_2^2(\rho_1, \hat{\rho}_1) \leqslant e^{1+2\int_0^1 |C_t| dt} \mathcal{L}_{\mathsf{LMD}}(\hat{X}). \tag{3.15}$$

*where $C_t$ is the constant that appears in Assumption 3.3.*

The proof is given in Appendix B. We now state an analogous result for the Eulerian case.

**Proposition 3.10** (Eulerian error bound). *Let $X_{s,t} : \mathbb{R}^d \to \mathbb{R}^d$ denote the flow map for $b$, and let $\hat{X}_{s,t} : \mathbb{R}^d \to \mathbb{R}^d$ denote an approximate flow map. Given $x_0 \sim \rho_0$, let $\hat{\rho}_1$ be the density of $\hat{X}_{0,1}(x_0)$ and let $\rho_1$ be the target density of $X_{0,1}(x_0)$. Then,*

$$W_2^2(\rho_1, \hat{\rho}_1) \leqslant e^1 \mathcal{L}_{\mathsf{EMD}}(\hat{X}). \tag{3.16}$$

The proof is also given in Appendix B. The result in Proposition 3.10 appears stronger than the result in Proposition 3.9, because it is independent of any Lipschitz constant. Notice, however, that unlike (3.15) the bound (3.16) involves the spatial gradient of the map $X_{s,t}$, which may be more difficult to control. In our numerical experiments, we found the best performance when using the Lagrangian distillation loss, rather than the Eulerian distillation loss. We hypothesize and provide numerical evidence that this originates from avoiding the spatial gradient present in the Eulerian distillation loss; in several cases of interest, the learned map can be singular or nearly singular, so that the spatial gradient is not well defined everywhere. This leads

---

**Algorithm 3:** Flow map matching (FMM)

---

**input:** Interpolant coefficients $\alpha_t, \beta_t, \gamma_t$; weight function $w_{s,t}$; batch size $M$.
**repeat**

> Draw batch $(s_i, t_i, x_0^i, x_1^i, z_i)_{i=1}^M$ from $w_{s,t} \times \rho(x_0, x_1) \times \mathsf{N}(z; 0, Id)$.
> Compute $I_{t_i} = \alpha_{t_i} x_0^i + \beta_{t_i} x_1^i + \gamma_{t_i} z_i$ and $\dot{I}_{t_i} = \dot{\alpha}_{t_i} x_0^i + \dot{\beta}_{t_i} x_1^i + \dot{\gamma}_{t_i} z_i$.
> Compute $\hat{\mathcal{L}}_{\mathrm{FMM}} = \frac{1}{M} \sum_{i=1}^M \left( |\partial_t \hat{X}_{s_i,t_i}(\hat{X}_{t_i,s_i}(I_{t_i})) - \dot{I}_{t_i}|^2 + |\hat{X}_{s_i,t_i}(\hat{X}_{t_i,s_i}(I_{t_i})) - I_{t_i}|^2 \right)$.
> Take gradient step on $\hat{\mathcal{L}}_{\mathrm{FMM}}$ to update $\hat{X}$.

**until** *converged*;
**output:** Flow map $\hat{X}$.

---

to training difficulties that manifest themselves as fuzzy boundaries on the checkerboard dataset and blurry images on image datasets.

### 3.5 Direct training with flow map matching (FMM)

We now give a loss function for *direct* training of the flow map that does not require a pre-trained $b$.

**Proposition 3.11** (Flow map matching). *The flow map is the global minimizer over $\hat{X}$ of the loss*

$$\mathcal{L}_{\mathsf{FMM}}(\hat{X}) = \int_{[0,1]^2} w_{s,t} \left( \mathbb{E}\big[ |\partial_t \hat{X}_{s,t}(\hat{X}_{t,s}(I_t)) - \dot{I}_t|^2 \big] + \mathbb{E}\big[ |\hat{X}_{s,t}(\hat{X}_{t,s}(I_t)) - I_t|^2 \big] \right) dsdt, \qquad (3.17)$$

*subject to the boundary condition $\hat{X}_{s,s}(x) = x$ for all $x \in \mathbb{R}^d$ and for all $s \in \mathbb{R}$. In (3.17), $w_{s,t} > 0$ and $\mathbb{E}$ is taken over the coupling $(x_0, x_1) \sim \rho(x_0, x_1)$ and $z \sim \mathsf{N}(0, Id)$.*

In the loss (3.17), we are free to adjust the weight factor $w_{s,t}$, as illustrated in Figure 2. However, since we need to learn both the map $X_{s,t}$ and its inverse $X_{t,s}$, it is necessary to enforce the symmetry property $w_{t,s} = w_{s,t}$. If we learn the map for all $(s, t) \in [0,1]^2$ using, for example, $w_{s,t} = 1$, then we can generate samples from $\rho_1$ in one step via $X_{0,1}(x_0)$ with $x_0 \sim \rho_0$. If we learn the map in a strip, as shown in Figure 2, the learning becomes simpler but we need to use multiple steps to generate samples from $\rho_1$; the number of steps then depends of the width of the strip. We note that the second term enforcing invertibility in (3.17) comes at no additional cost on the forward pass, because $\hat{X}_{s,t}(\hat{X}_{t,s}(I_t))$ can be computed at the same time as $\partial_t \hat{X}_{s,t}(\hat{X}_{t,s}(I_t))$ with standard Jacobian-vector product functionality in modern deep learning packages. A summary of the flow map matching procedure is given in Algorithm 3.

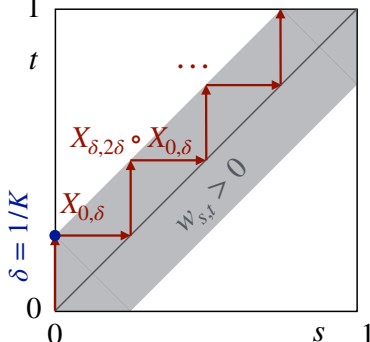

**Figure 2:** Schematic illustrating the weight $w_{s,t}$ in the FMM loss, which can be tuned to arrive at different learning schemes.

### 3.6 Eulerian estimation or Eulerian distillation?

In light of Proposition 3.11, the reader may wonder whether we could also perform direct estimation in the Eulerian setup, using for example the objective

$$\int_{[0,1]^2} w_{s,t} \mathbb{E}\Big[ \big| \partial_s \hat{X}_{s,t}(I_s) + \dot{I}_s \cdot \nabla \hat{X}_{s,t}(I_s) \big|^2 \Big] dsdt. \qquad (3.18)$$

The above loss may be obtained from (3.13) by replacing the appearance of $b_s(I_s)$ with $\dot{I}_s$. Unfortunately, (3.18) is not equivalent to (3.13). To see why, we can expand the expectation in (3.18):

$$\mathbb{E}\Big[ \big| \partial_s \hat{X}_{s,t}(I_s) + \dot{I}_s \cdot \nabla \hat{X}_{s,t}(I_s) \big|^2 \Big]$$
$$= \mathbb{E}\Big[ \big| \partial_s \hat{X}_{s,t}(I_s) \big|^2 + 2(\dot{I}_s \cdot \nabla \hat{X}_{s,t}(I_s)) \cdot \partial_s \hat{X}_{s,t}(I_s) + \big| \dot{I}_s \cdot \nabla \hat{X}_{s,t}(I_s) \big|^2 \Big]. \qquad (3.19)$$

---

**Algorithm 4:** Progressive flow map matching (PFMM)

---

**input:** Interpolant coefficients $\alpha_t, \beta_t, \gamma_t$; weight $w_{s,t}$; $K$-step flow map $\hat{X}$; batch size $M$.

**repeat**

    Draw batch $(s_i, t_i, I_{s_i})_{i=1}^M$ and compute $t_k^i = s_i + (k-1)(t_i - s_i)$ for $k = 1, \ldots, K$.

    Compute $\hat{\mathcal{L}}_{\mathsf{PFMM}} = \frac{1}{M} \sum_{i=1}^M \left( |\check{X}_{s_i,t_i}(I_{s_i}) - \left( \hat{X}_{t_{K-1}^i, t_K^i} \circ \cdots \circ \hat{X}_{t_1^i, t_2^i} \right)(I_{s_i})|^2 \right)$.

    Take gradient step on $\hat{\mathcal{L}}_{\mathsf{PFMM}}$ to update $\check{X}$.

**until** *converged*;

**output:** One-step flow map $\check{X}$.

---

The cross term is linear in $\dot{I}_s$, so that we can use the tower property of the conditional expectation to see

$$\mathbb{E}\left[ (\dot{I}_s \cdot \nabla \hat{X}_{s,t}(I_s)) \cdot \partial_s \hat{X}_{s,t}(I_s) \right] = \mathbb{E}\left[ (\mathbb{E}[\dot{I}_s | I_s] \cdot \nabla \hat{X}_{s,t}(I_s)) \cdot \partial_s \hat{X}_{s,t}(I_s) \right],$$
$$= \mathbb{E}\left[ (b_s(I_s) \cdot \nabla \hat{X}_{s,t}(I_s)) \cdot \partial_s \hat{X}_{s,t}(I_s) \right]. \quad (3.20)$$

However, the tower property cannot be applied to the last term in (3.19) since it is quadratic in $\dot{I}_s$, i.e.

$$\mathbb{E}\left[ |\dot{I}_s \cdot \nabla \hat{X}_{s,t}(I_s)|^2 \right] \neq \mathbb{E}\left[ |b_s(I_s) \cdot \nabla \hat{X}_{s,t}(I_s)|^2 \right]. \quad (3.21)$$

Since this term depends on $\hat{X}$, it cannot be neglected in the minimization, and the minimizer of (3.18) is not the same as that of (3.13). Recognizing this difficulty, consistency models (Song et al., 2023; Song and Dhariwal, 2023; Kim et al., 2024a) place a stopgrad$(\cdot)$ on the term $\dot{I}_s \cdot \nabla \hat{X}_{s,t}(I_s)$ when computing the gradient of the loss (3.18). The resulting iterative scheme used to update $\hat{X}$ then has the correct fixed point at $\hat{X} = X$, as summarized in the following proposition.

**Proposition 3.12** (Eulerian estimation). *The flow map $X_{s,t}$ is a critical point of the loss*

$$\mathcal{L}_{\mathsf{EE}}(\hat{X}) = \int_{[0,1]^2} w_{s,t} \mathbb{E}\left[ |\partial_s \hat{X}_{s,t}(I_s) - \mathsf{stopgrad}(\dot{I}_s \cdot \nabla \hat{X}_{s,t}(I_s))|^2 \right] ds dt, \quad (3.22)$$

*subject to the boundary condition $\hat{X}_{s,s}(x) = x$ for all $x \in \mathbb{R}^d$ and for all $s \in \mathbb{R}$. In (3.22), $w_{s,t} > 0$ and $\mathbb{E}$ is taken over the coupling $(x_0, x_1) \sim \rho(x_0, x_1)$, $z \sim \mathsf{N}(0, Id)$. The operator $\mathsf{stopgrad}(\cdot)$ indicates that the argument is ignored when computing the gradient.*

We note that it is challenging to guarantee that the critical point described in Proposition 3.12 is stable and attractive for an iterative gradient-based scheme, as the objective function is not guaranteed to decrease due to the appearance of the stopgrad$(\cdot)$ operator. Nonetheless, some works have found success with approaches related to this objective on large-scale image generation problems (Song et al., 2023).

### 3.7 Progressive distillation

Empirically, we found directly learning a one-step map to be challenging in practice. Convergence was significantly improved by taking $w_{s,t} = w_{t,s} = \mathbb{I}(|t - s| \leqslant 1/K)$ for some $K \in \mathbb{N}$ where $\mathbb{I}$ denotes an indicator function. Given such a $K$-step model, it can be converted into a one-step model using a map distillation loss that is similar to progressive distillation (Salimans and Ho, 2022) and neural operator approaches (Zheng et al., 2023).

**Lemma 3.13** (Progressive flow map matching). *Let $\hat{X}$ be a two-time flow map. Given $K \in \mathbb{N}$, let $t_k = s + \frac{k-1}{K-1}(t - s)$ for $k = 1, \ldots, K$. Then the unique minimizer over $\check{X}$ of the objective*

$$\mathcal{L}_{\mathsf{PFMM}}(\check{X}) = \int_{[0,1]^2} w_{s,t} \mathbb{E}\left[ |\check{X}_{s,t}(I_s) - \left( \hat{X}_{t_{K-1}, t_K} \circ \cdots \circ \hat{X}_{t_1, t_2} \right)(I_s)|^2 \right] ds dt, \quad (3.23)$$

*produces the same output in one step as the $K$-step iterated map $\hat{X}$. Here $w_{s,t} > 0$, and $\mathbb{E}$ is taken over the coupling $(x_0, x_1) \sim \rho(x_0, x_1)$ and $z \sim \mathsf{N}(0, Id)$.*

We note that $\hat{X}$ is fixed in (3.23) and serves as the teacher, so we only need to compute the gradient with respect to the parameters of $\check{X}$. In practice, we may train $\hat{X}$ using (3.17) over a class of neural networks and then freeze its parameters. We may then use (3.23) to distill $\hat{X}$ into a more efficient model $\check{X}$, which can be initialized from the parameters of $\hat{X}$ for an efficient warm start. Alternatively, we can add (3.23) to (3.17) and train a single network $\hat{X}$ by setting $\check{X} = \hat{X}$ in (3.23) and placing a stopgrad($\cdot$) on the multistep term.

If $K$ evaluations of $\hat{X}$ are expensive, we may iteratively minimize (3.23) with some number $M < K$ evaluations of $\hat{X}$ and then replace $\hat{X}$ by $\check{X}$, similar to progressive distillation (Salimans and Ho, 2022). For example, we may take $M = 2$ and then minimize (3.23) $\lceil \log_2 K \rceil$ times to obtain a one-step map. Alternatively, we can first generate a dataset of $(s, t, I_s, (\hat{X}_{t_{K-1}, t_K} \circ \cdots \circ \hat{X}_{t_1, t_2})(I_s))$ in a parallel offline phase, which converts (3.23) into a simple least-squares problem. Finally, if we are only interested in using the map forward in time, we can set $w_{s,t} = 1$ if $s \leqslant t$ and $w_{s,t} = 0$ otherwise. The resulting procedure is summarized in Algorithm 4.

## 4 Numerical Realizations

In this section, we study the efficacy of the four methods introduced in Section 3: the Lagrangian map distillation discussed in Corollary 3.7, the Eulerian map distillation discussed in Corollary 3.8, the direct training approach of Proposition 3.11, and the progressive flow map matching approach of Lemma 3.13. We consider their performance on a two-dimensional checkerboard dataset, as well as in the high-dimensional setting of image generation, to highlight differences in their training efficiency and performance.

To ensure that the boundary conditions on the flow map $\hat{X}_{s,t}$ defined in (3.7) are enforced, in all experiments, we parameterize the map using the ansatz

$$X_{s,t}(x) = x + (t - s)v^{\theta}_{s,t}(x), \tag{4.1}$$

where $v^{\theta}_{s,t}(x) : [0, T]^2 \times \mathbb{R}^d \to \mathbb{R}^d$ is a neural network with parameters $\theta$.

### 4.1 2D Illustration

As a simple illustration of our method, we consider learning the flow map connecting a two-dimensional Gaussian distribution to the checkerboard distribution presented in Figure 3. Note that this example is challenging because the target density is supported on a compact set, and because the target is discontinuous at the edge of this set. This mapping can be achieved, as discussed in Section 3, in various ways: (a) implicitly, by solving (3.2) with a learned velocity field using stochastic interpolants (or a diffusion model), (b) directly, using the flow map matching objective in (3.17), (c) progressively matching the flow map using (3.23), or (d/e) distilling the map using the Eulerian (3.13) or Lagrangian (3.11) losses. In each case, we use a fully connected neural network with 512 neurons per hidden layer and 6 layers to parameterize either a velocity field $\hat{b}_t(x)$ or a flow map $\hat{X}_{s,t}(x)$. We optimize each loss using the Adam (Kingma and Ba, 2017) optimizer for $5 \times 10^4$ training iterations. The results are presented in Figure 3, where we observe that using the one-step $\hat{X}_{0,1}(x)$ directly learned by minimizing (3.17) over the entire interval $(s, t) \in [0, 1]^2$ performs worse than learning with $|t - s| < 0.25$ and sampling with 4 steps. With this in mind, we use the 4-step map as a teacher to minimize the PFMM loss, which produces a high-performing one-step map. We also note that the EMD loss performs worse than the LMD loss when distilling the map from a learned velocity field.

### 4.2 Image Generation

Motivated by the above results, we consider a series of image generation experiments on the CIFAR-10 and ImageNet-$32 \times 32$ datasets. For comparison, we benchmark the method against alternative techniques that seek to lower the number of steps needed to produce samples with stochastic interpolant models, e.g. by straightening the ODE trajectories using minibatch OT (Pooladian et al., 2023; Tong et al., 2023). We train all of our models from scratch, so as to control the design space of the comparison. For clarity, we label when benchmark numbers are quoted from the literature.

For learning of the flow map, we use a U-Net architecture following (Dhariwal and Nichol, 2021). For LMD and EMD that require a pre-trained velocity field to distill, we also use a U-Net architecture for $b$. Because the

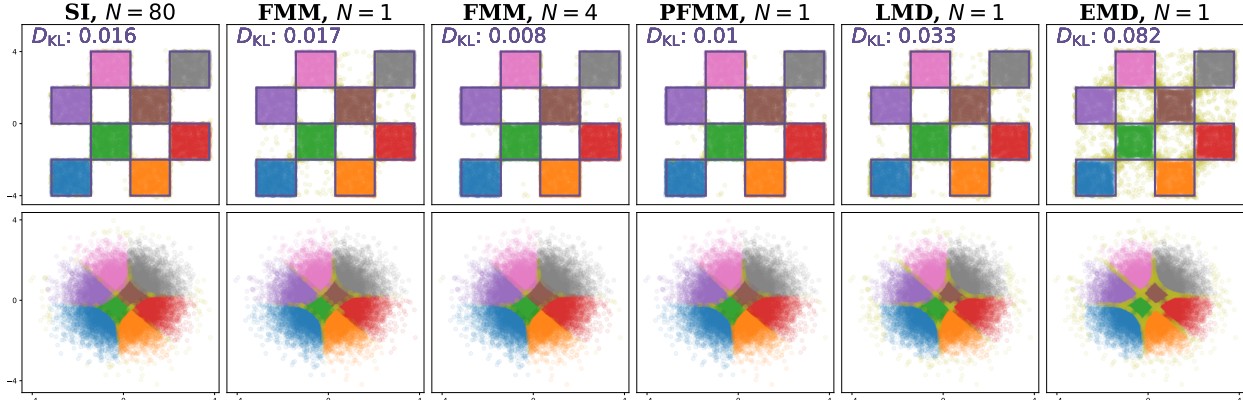

**Figure 3: Two-dimensional results.** Comparison of the various map-matching procedures on the 2D checkerboard dataset, with the results from the probability flow ODE of a stochastic interpolant integrated using $N = 80$ discretization steps as reference (top, first panel from left). The one-step map obtained by FMM when learning on $(s, t) = [0, 1]^2$ (top, second panel) performs as well as SI. Moreover, the accuracy improves if we allow four steps instead of one (top, third panel). This four-step map can be accurately distilled into a one-step map via PFMM (top, fourth panel). The one-step map obtained by distilling the pre-trained $b$ via LMD (top, fifth panel) performs reasonably well too, and is better than the one-step map obtained by distilling the same $b$ via EMD (top, sixth panel). A KL-divergence between each model distribution and the target is provided to quantify performance, indicating that FMM, its progressive distillation, and LMD are closest to the probability flow ODE baseline. The bottom row indicates, by color, how points from the Gaussian base are assigned by each of the respective maps. The yellow dots are points that mistakenly land outside the checkerboard. These results indicate that the primary source of error in each case is handling the discontinuity of the optimal map at the edges of the checker. See Appendix D for more details.

flow map $X_{s,t}$ is a function of two times, we modify the architecture. Both $s$ and $t$ are embedded identically to $t$ in the original architecture. The result is concatenated and treated like $t$ in the original architecture for downstream layers. We benchmark the performance of the methods using the Frechet Inception Distance (FID), which computes a measure of similarity between real images from the dataset and those generated by our models. In addition, we compute what we denote as the Teacher-FID (T-FID). This metric computes the same measure of similarity, but now between images generated by the teacher model and those generated by

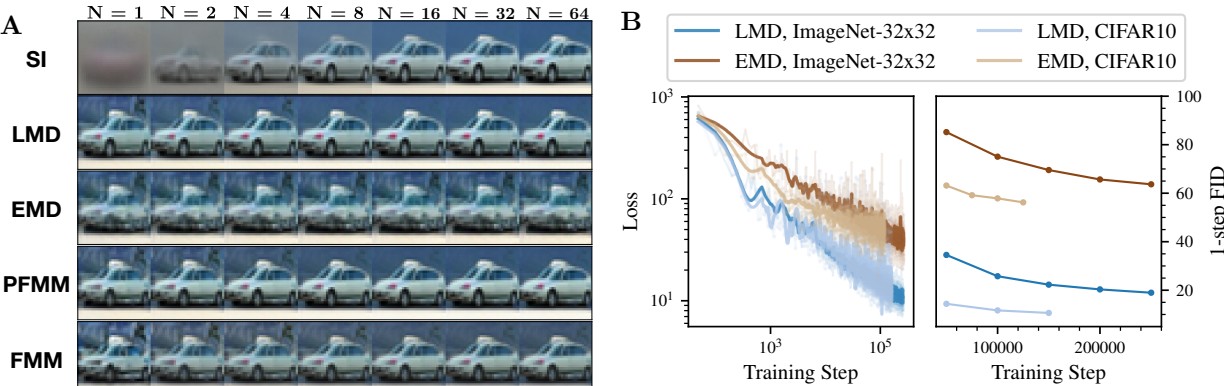

**Figure 4:** (A) Qualitative comparison between the standard stochastic interpolant approach (SI), Lagrangian map distillation (LMD), Eulerian map distillation (EMD), and progressive flow map matching (PFMM). SI produces good images for a sufficiently large number of steps, but performs poorly for few steps. LMD performs well in the very-few step regime, and outperforms EMD significantly. PFMM performs well at any number of steps, though performs slightly worse than LMD in the very-few step regime. (B) Quantitative comparison between EMD and LMD on both CIFAR-10 and ImageNet $32 \times 32$. Despite both having the same minimizer, LMD trains faster, and attains a lower loss value and a lower FID for a fixed number of training steps.

| Method | N=2 | | N=4 | | Baseline |
|---|---|---|---|---|---|
| | FID | T-FID | FID | T-FID | |
| SI | 112.42 | - | 34.84 | - | 5.53 |
| EMD | 48.32 | 34.19 | 44.35 | 30.74 | 5.53 |
| LMD | 7.13 | 1.27 | 6.04 | 1.05 | 5.53 |
| PFMM | 18.35 | 7.02 | 11.14 | 1.52 | 8.44 |

**Table 1:** Comparison of various distillation methods using FID and Teacher-FID metrics on the CIFAR-10 dataset. Note that for PFMM, no velocity model (e.g. from a stochastic interpolant) is needed. It relies solely on the minimization of (3.17) and (3.23). Baseline indicates the FID of the teacher model (a velocity field for EMD and LMD integrated with an adaptive fifth-order Runge-Kutta scheme, and a flow map for PFMM) against the true data.

the distilled model, rather than leveraging the original dataset. This measure allows us to directly benchmark the convergence of the distillation method, as it captures discrepancies between the distribution of samples generated by the teacher and the distribution of samples generated by the student. In addition, this allows us to benchmark accuracy independent of the overall performance of the teacher, as our teacher models were trained with limited compute.

**Sampling efficiency**   In Table 1, we compute the FID and T-FID for the stochastic interpolant, Eulerian, Lagrangian, and progressive distillation models on 2 and 4-step generation for CIFAR-10. The stochastic interpolant was trained to a baseline FID (sampling with an adaptive solver) of 5.53, and was used as the teacher for EMD and LMD. The teacher for PFMM was an FMM model trained with $|t - s| < 0.25$ to an FID of 8.44 using 8-step sampling. We observe that LMD and EMD methods can effectively distill their teachers and obtain low T-FID scores. In addition, the 2 and 4-step samples from these methods far outperform the stochastic interpolant. This sampling efficiency is also apparent in the left side of Figure 4, in which with just 1 to 4 steps, the LMD and PFMM methods can produce effective samples, particularly when compared to the flow matching approach.

Without any distillation, FMM can also produce effective few-step maps. Training an FMM model on the ImageNet-32 $\times$ 32 dataset, we observe (Table 2) that FMM achieves much better few-step FID when compared to denoising diffusion models (DDPM), and better FID than mini-batch OT interpolant methods (Pooladian et al., 2023). In the higher-step regime, the interpolant methods perform marginally better.

| N | DDPM | BatchOT | FMM (Ours) |
|---|---|---|---|
| 20 | 63.08 | **7.71** | 9.68 |
| 8 | 232.97 | 15.64 | **12.61** |
| 6 | 275.28 | 22.08 | **14.48** |
| 4 | 362.37 | 38.86 | **16.90** |

**Table 2:** FID scaling with number of function evaluations $N$ to produce a sample on ImageNet-32 $\times$ 32. We show a comparison between DDPM (Ho et al., 2020) and multi-sample Flow Matching using the BatchOT method (Pooladian et al., 2023) to flow map matching. The first two columns are quoted from Pooladian et al. (2023). Note that no distillation is used here, but rather direct minimization of (3.17), using $|t - s| < 0.25$.

**Eulerian vs Lagrangian distillation**   Remarkably, we find a stark performance gap between the Eulerian and Lagrangian distillation schemes. This is evident in both parts of Figure 4, where we see that higher-step sampling with EMD only marginally improves image quality, and where the LMD loss for both CIFAR10 and ImageNet-32 $\times$ 32 converges an order of magnitude faster than the EMD loss. The same holds for FIDs over training, given in the right-most plot in the figure. Note that both LMD and EMD loss functions have a global minimum at 0, so that the loss plots suggest continued training will improve distillation quality, but at very different rates.

### 4.2.1  Efficient Style Transfer

To illustrate some of the downstream tasks facilitated by our approach, we now describe a means of performing style transfer with the two-time invertible flow map that we call "consistency style transfer". A class conditional sample $x_1$ with class label $y$ can be partially inverted to an earlier time $s'$ via $X_{1,s'}(x_1; y)$ for $s' < 1$. From

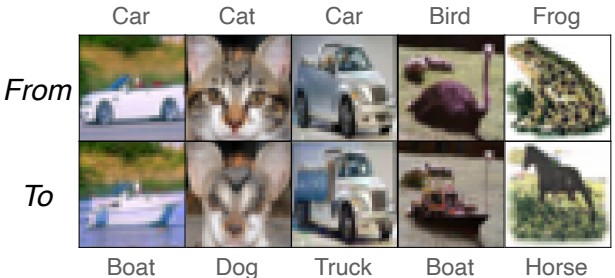

**Figure 5:** Consistency style transfer on CIFAR-10. Original class-conditional images from the dataset (top row) are pushed backward in time to $X_{1,s'=0.3}(x_1; y)$ and then pushed forward using a new class. The original styles of the images are maintained, while their subjects are replaced with those from the new class (bottom row).

here, replacing the class label with $y' \neq y$, we can resample the flow map $X_{s',1}(X_{1,s'}(x_1; y); y')$ to sample the conditional distribution associated to $y'$ while maintaining the style of the original class. We demonstrate this principle in Figure 5, which shows that maps discretized with $n = 8$ step can be used convert between classes. This serves both as verification of the cycle consistency as well as an illustration of the potential applications of the two-time flow map. In practice, the extent of the preserved style depends on how far back in time towards the Gaussian base at $s' = 0$ we push the original sample. Here, we use $s' = 0.3$.

## 5  Conclusion

In this work, we developed a framework for learning the two-time flow map for generative modeling: either by distilling a pre-trained velocity model with the Eulerian or Lagrangian losses, or by directly training in the stochastic interpolant framework. We empirically observe that while using more steps with the learned map improves sample quality, a substantially lower number is needed when compared to other generative models built on dynamical transport. Future work will investigate how to improve the training and the neural network architecture so as to further reduce the number of steps without sacrificing accuracy, and to improve convergence for direct training of one-step maps.

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

## A  Probability flow ODEs from stochastic interpolants and score-based diffusion models

For the reader's convenience, we now recall how to construct probability flow ODEs using either flow matching with stochastic interpolants or score-based diffusion models. We also recall the connection between these two formalisms.

### A.1  Transport equation

Generative models based on probability flow ODEs leverage the property that solutions to (3.2) push forward their initial conditions onto samples from the target:

**Lemma A.1** (Transport equation). *Let $\rho_t = \mathrm{Law}(x_t)$ be the PDF of the solution to (3.2) assuming that $x_0 \sim \rho_0$. Then $\rho_t$ satisfies*

$$\partial_t \rho_t(x) + \nabla \cdot (b_t(x)\rho_t(x)) = 0, \qquad \rho_{t=0}(x) = \rho_0(x) \tag{A.1}$$

*where $\nabla$ denotes a gradient with respect to $x$.*

The interest of this result is that it can be used in reverse: if we show that a PDF $\rho_t$ satisfies the transport equation (A.1), then in practice we can use the probability flow ODE (3.2) to sample $\rho_t$ at any time $t > 0$, so long as we can sample initial conditions from $\rho_0$.

*Proof.* The proof proceeds via the weak formulation of (A.1). Let $\phi \in C_b^1(\mathbb{R}^d)$ denote an arbitrary test function. By definition,

$$\forall t \in [0,1] \quad : \quad \int_{\mathbb{R}^d} \phi(x)\rho_t(x)dx = \mathbb{E}[\phi(x_t)] \tag{A.2}$$

where $x_t$ is given by (3.1) and where the expectation on the right hand-side is taken over the law of the initial conditions $x_{t=0} = x_0 \sim \rho_0$. Taking the time derivative of this equality, we deduce that

$$\begin{aligned}
\int_{\mathbb{R}^d} \phi(x)\partial_t \rho_t(x)dx &= \mathbb{E}[\dot{x}_t \cdot \nabla\phi(x_t)] && \text{by the chain rule} \\
&= \mathbb{E}[b_t(x_t) \cdot \nabla\phi(x_t)] && \text{using the ODE (3.2)} \\
&= \int_{\mathbb{R}^d} b_t(x) \cdot \nabla\phi(x)\rho_t(x)dx && \text{by definition of } \rho_t(x)
\end{aligned} \tag{A.3}$$

This is the weak form of the transport equation (A.1). It can be written as (A.1), since it admits strong solutions by our assumptions on $b_t(x)$. $\square$

### A.2  Stochastic interpolants and probability flows

The stochastic interpolant framework offers a simple and versatile paradigm to construct generative models.

**Definition 3.1** (Stochastic Interpolant). *The stochastic interpolant $I_t$ between probability densities $\rho_0$ and $\rho_1$ is the stochastic process given by*

$$I_t = \alpha_t x_0 + \beta_t x_1 + \gamma_t z, \tag{3.1}$$

*where $\alpha, \beta, \gamma^2 \in C^1([0,1])$ satisfy $\alpha_0 = \beta_1 = 1$, $\alpha_1 = \beta_0 = 0$, and $\gamma_0 = \gamma_1 = 0$. In (3.1), $(x_0, x_1)$ is drawn from a coupling $(x_0, x_1) \sim \rho(x_0, x_1)$ that satisfies the marginal constraints $\int_{\mathbb{R}^d} \rho(x_0, x_1)dx_0 = \rho_1(x_1)$ and $\int_{\mathbb{R}^d} \rho(x_0, x_1)dx_1 = \rho_0(x_0)$. Moreover, $z \sim \mathsf{N}(0, Id)$ with $z \perp (x_0, x_1)$.*

**Proposition 3.2** (Probability Flow). *For all $t \in [0,1]$, the probability density of $I_t$ is the same as the probability density of the solution to*

$$\dot{x}_t = b_t(x_t), \qquad x_{t=0} = x_0 \sim \rho_0, \tag{3.2}$$

*where $b : [0,1] \times \mathbb{R}^d \to \mathbb{R}^d$ is the time-dependent velocity field (or drift) given by*

$$b_t(x) = \mathbb{E}[\dot{I}_t | I_t = x]. \tag{3.3}$$

*In (3.3), $\mathbb{E}[\cdot | I_t = x]$ denotes an expectation over the coupling $(x_0, x_1) \sim \rho(x_0, x_1)$ and $z \sim \mathsf{N}(0, Id)$ conditioned on the event $I_t = x$.*

*Proof.* Let $\phi \in C_b^1(\mathbb{R}^d)$ denote an arbitrary test function. By definition,

$$\forall t \in [0,1] \quad : \quad \int_{\mathbb{R}^d} \phi(x) \rho_t(x) dx = \mathbb{E}[\phi(I_t)] \tag{A.4}$$

where $I_t$ is given by (3.1) and where the expectation on the right-hand side is taken over the coupling $(x_0, x_1) \sim \rho(x_0, x_1)$ and $z \sim \mathsf{N}(0, I)$. Taking the time derivative of this equality, we deduce that

$$
\begin{aligned}
\int_{\mathbb{R}^d} \phi(x) \partial_t \rho_t(x) dx &= \mathbb{E}[\dot{I}_t \cdot \nabla \phi(I_t)] && \text{by the chain rule} \\
&= \mathbb{E}[\mathbb{E}[\dot{I}_t | I_t] \cdot \nabla \phi(I_t)] && \text{by the tower property of the conditional expectation} \\
&= \mathbb{E}[b_t(I_t) \cdot \nabla \phi(I_t)] && \text{by definition of } b_t(x) \text{ in (3.3)} \\
&= \int_{\mathbb{R}^d} b_t(x) \cdot \nabla \phi(x) \rho_t(x) dx && \text{by definition of } \rho_t(x)
\end{aligned}
\tag{A.5}
$$

This is the weak form of the transport equation (A.1). $\qquad\square$

This result implies that the PDF $\rho_t$ of the stochastic interpolant $I_t$ is also the PDF of the solution $x_t$ of the probability flow ODE (3.2) with the velocity field $b_t(x)$ defined in (3.3). Algorithmically, this means that we can use the associated flow map as a generative model.

### A.3 Score-based diffusion models

For simplicity, we focus on variance-preserving diffusions; extensions to more general classes of diffusions are straightforward. These models are based on variants of the Ornstein-Uhlenbeck process, defined as the solution to the stochastic differential equation (SDE):

$$dX_t = -X_t dt + \sqrt{2}\, dW_t, \qquad X_{t=0} = a \sim \rho_*, \tag{A.6}$$

where $W_t$ is a Wiener process. The solution to (A.6) for the initial condition $X_{t=0} = a$ is:

$$X_t = a\, e^{-t} + \sqrt{2} \int_0^t e^{-t+t'} dW_{t'}. \tag{A.7}$$

By the Itô Isometry, the second term on the right-hand side is a Gaussian process with mean zero and covariance given by

$$\mathbb{E}\left[\left(\sqrt{2} \int_0^t e^{-t+t'} dW_{t'}\right)^2\right] = 2Id \int_0^t e^{-2t+2t'} dt' = (1 - e^{-2t})Id. \tag{A.8}$$

Therefore, at any time $t$, the law of (A.7) is that of a Gaussian random variable with mean $xe^{-t}$ and covariance $(1 - e^{-2t})Id$. That is, it can be represented as

$$X_t \overset{d}{=} a\, e^{-t} + \sqrt{1 - e^{-2t}} z, \tag{A.9}$$

where $a \sim \rho_*$, $z \sim N(0, Id)$, and $a \perp z$. (A.9) shows that the law of $X_t$ converges exponentially fast to that of the standard normal variable as $t \to \infty$ for all initial conditions $a$.

The key insight of score-based diffusion models is to time-reverse the SDE (A.6) to obtain a process that turns Gaussian samples into samples from the target. Alternatively, this time-reversal can be performed using the associated probability flow ODE. To do so, we can start from the evolution equation for the PDF of the solution to the SDE (A.6). Denoting this PDF by $\tilde{\rho}_t$, it satisfies the Fokker-Planck equation (FPE)

$$\partial_t \tilde{\rho}_t = \nabla \cdot (x \tilde{\rho}_t) + \Delta \tilde{\rho}_t, \qquad \tilde{\rho}_0 = \rho_*. \tag{A.10}$$

To time-reverse the solution to (A.10) we define $\rho_t = \tilde{\rho}_{T-t}$ for some large $T > 0$, and derive an equation for $\rho_t$ from (A.10):

$$\begin{aligned}
\partial_t \tilde{\rho}_t &= -\partial_t \tilde{\rho}_{T-t} \\
&= -\nabla \cdot (x \tilde{\rho}_{T-t}) - \Delta \tilde{\rho}_{T-t} \\
&= -\nabla \cdot ([x + \nabla \log \tilde{\rho}_{T-t}] \tilde{\rho}_{T-t}) \\
&= -\nabla \cdot ([x + \nabla \log \rho_t] \rho_t)
\end{aligned} \tag{A.11}$$

where we used the identity $\Delta \tilde{\rho}_{T-t} = \nabla \cdot (\tilde{\rho}_{T-t} \nabla \log \tilde{\rho}_{T-t})$. Equation (A.11) can be written as

$$\partial_t \tilde{\rho}_t + \nabla \cdot ([x + s_t(x)] \rho_t) = 0, \qquad \rho_{t=0} = \tilde{\rho}_{t=T} \approx N(0, Id) \tag{A.12}$$

where $s_t(x) = \nabla \log \rho_t(x) = \nabla \log \tilde{\rho}_{T-t}(x)$ is the score. This quantity can be estimated by regression using the solutions to (A.6), which are explicitly available via (A.9). Equation (A.12) is in the form of (A.1) with a velocity field $b_t(x)$ given by

$$b_t(x) = x + s_t(x). \tag{A.13}$$

Flow map matching can therefore be used to distill score-based diffusion models using this velocity field.

### A.4 Connections

Here we show how score-based diffusion models can be related to stochastic interpolants. Recall that the law of the solution to the SDE (A.6) is given by (A.9). If we change time according to $t \mapsto -\log t$, we map $[0, \infty)$ onto $[1, 0)$ and we arrive at

$$X_{-\log t} \overset{d}{=} I_t = at + \sqrt{1 - t^2}\, z, \qquad a \sim \rho_*, \;\; z \sim N(0, Id), \;\; \text{and} \;\; a \perp z. \tag{A.14}$$

This is a valid stochastic interpolant if we set $a = x_1$, $\beta_t = t$, $\alpha_t = 0$, and $\gamma_t = \sqrt{1 - t^2}$ in (3.1)[3]. Hence, we have shown that the probability flow of the score-based diffusion model with velocity field (A.13) can also be studied by using the stochastic interpolant (A.14). Note that this reformulation has the advantage that it eliminates the small bias incurred by the finite value of $T$ in score-based diffusion models, since the stochastic interpolant (A.14) transports the Gaussian random variable $z$ at time $t = 0$ onto the data point $a$ at time $t = 1$ exactly.

## B  Proofs for Section 3

**Proposition 3.5.** *The flow map $X_{s,t}(x)$ is the unique solution to the Lagrangian equation*

$$\partial_t X_{s,t}(x) = b_t(X_{s,t}(x)), \qquad X_{s,s}(x) = x, \tag{3.7}$$

*for all $(s, t, x) \in [0, 1]^2 \times \mathbb{R}^d$. In addition, it satisfies*

$$X_{t,\tau}(X_{s,t}(x)) = X_{s,\tau}(x) \tag{3.8}$$

*for all $(s, t, \tau, x) \in [0, 1]^3 \times \mathbb{R}^d$. In particular $X_{s,t}(X_{t,s}(x)) = x$ for all $(s, t, x) \in [0, 1]^2 \times \mathbb{R}^d$, i.e. the flow map is invertible.*

---

[3]Note that $\gamma_0 = 1$ here, consistent with the fact that the base PDF is Gaussian, so that $x_0$ and $z$ can be lumped into a single Gaussian variable

*Proof.* Taking the derivative with respect to $t$ of $X_{s,t}(x_s) = x_t$, we deduce

$$\partial_t X_{s,t}(x_s) = \dot{x}_t = b_t(x_t) = b_t(X_{s,t}(x_s)) \tag{B.1}$$

where we used the ODE (3.2) to obtain the second equality. Conversely, since the solutions to (3.2) and (3.7) from a given initial condition are unique, if we solve (3.7) on $[s,t]$ with the condition $X_{s,s}(x_s) = x_s$ we must have $X_{s,t}(x_s) = x_t$ for all $(s,t) \in [0,1]^2$. Evaluating this expression at $x_s = x$ gives (3.7). Also, for all $(s,t,t) \in [0,T]^3$, we have

$$X_{t,t}(X_{s,t}(x_s)) = X_{t,t}(x_t) = x_t = X_{s,t}(x_s). \tag{B.2}$$

Evaluating this expression at $x_s = x$ gives (3.8). $\qquad\square$

**Corollary 3.7** (Lagrangian map distillation). *Let $w_{s,t} \in L^1([0,1]^2)$ be a weight function satisfying $w_{s,t} > 0$ and let $I_s$ be the stochastic interpolant defined in (3.1). Then the flow map is the global minimizer over $\hat{X}$ of the loss*

$$\mathcal{L}_{\mathsf{LMD}}(\hat{X}) = \int_{[0,1]^2} w_{s,t} \mathbb{E}\big[|\partial_t \hat{X}_{s,t}(I_s) - b_t(\hat{X}_{s,t}(I_s)|^2\big] \, ds dt, \tag{3.11}$$

*subject to the boundary condition that $\hat{X}_{s,s}(x) = x$ for all $x \in \mathbb{R}^d$ and $s \in [0,1]$. In (3.11), $\mathbb{E}$ denotes an expectation over the coupling $(x_0, x_1) \sim \rho(x_0, x_1)$ and $z \sim \mathsf{N}(0, Id)$.*

*Proof.* Equation (3.11) is a physics-informed neural network (PINN) (Raissi et al., 2019) loss that is minimized only when the integrand is zero, i.e., when (3.7) holds. $\qquad\square$

**Proposition 3.6.** *The flow map $X_{s,t}$ is the unique solution of the Eulerian equation,*

$$\partial_s X_{s,t}(x) + b_s(x) \cdot \nabla X_{s,t}(x) = 0, \qquad X_{t,t}(x) = x, \tag{3.10}$$

*for all $(s,t,x) \in [0,1]^2 \times \mathbb{R}^d$.*

*Proof.* Taking the derivative with respect to $s$ of $X_{s,t}(X_{t,s}(x)) = x$ and using the chain rule, we deduce that

$$\begin{aligned} 0 = \frac{d}{ds} X_{s,t}(X_{t,s}(x)) &= \partial_s X_{s,t}(X_{t,s}(x)) + \partial_s X_{t,s}(x) \cdot \nabla X_{s,t}(X_{t,s}(x)) \\ &= \partial_s X_{s,t}(X_{t,s}(x)) + b_s(X_{t,s}(x)) \cdot \nabla X_{s,t}(X_{t,s}(x)) \end{aligned} \tag{B.3}$$

where we used (3.7) to get the last equality. Evaluating this expression at $X_{t,s}(x) = y$, then changing $y$ into $x$, gives (3.10). $\qquad\square$

**Corollary 3.8** (Eulerian map distillation). *Let $w_{s,t}$ and $I_s$ be as in Corollary 3.7. Then the flow map is the global minimizer over $\hat{X}$ of the loss*

$$\mathcal{L}_{\mathsf{EMD}}(\hat{X}) = \int_{[0,1]^2} w_{s,t} \mathbb{E}\big[|\partial_s \hat{X}_{s,t}(I_s) + b_s(I_s) \cdot \nabla \hat{X}_{s,t}(I_s)|^2\big] \, ds dt, \tag{3.13}$$

*subject to the boundary condition $\hat{X}_{s,s}(x) = x$ for all $x \in \mathbb{R}^d$ and for all $s \in \mathbb{R}$.*

*Proof.* Equation (3.13) is a PINN loss that is minimized only when the integrand is zero, i.e, when (3.10) holds. $\qquad\square$

**Proposition 3.9** (Lagrangian error bound). *Let $X_{s,t} : \mathbb{R}^d \to \mathbb{R}^d$ denote the flow map for $b$, and let $\hat{X}_{s,t} : \mathbb{R}^d \to \mathbb{R}^d$ denote an approximate flow map. Given $x_0 \sim \rho_0$, let $\hat{\rho}_1$ be the density of $\hat{X}_{0,1}(x_0)$ and let $\rho_1$ be the target density of $X_{0,1}(x_0)$. Then,*

$$W_2^2(\rho_1, \hat{\rho}_1) \leqslant e^{1 + 2\int_0^1 |C_t| dt} \mathcal{L}_{\mathsf{LMD}}(\hat{X}). \tag{3.15}$$

*where $C_t$ is the constant that appears in Assumption 3.3.*

*Proof.* First observe that, by the one-sided Lipschitz condition (3.5),

$$
\begin{aligned}
\partial_t |X_{s,t}(x) - X_{s,t}(y)|^2 &= 2(X_{s,t}(x) - X_{s,t}(y)) \cdot (b_t(X_{s,t}(x)) - b_t(X_{s,t}(y))), \\
&\leqslant 2C_t |X_{s,t}(x) - X_{s,t}(y)|^2.
\end{aligned}
\tag{B.4}
$$

Equation (B.4) highlights that (3.5) gives a bound on the spread of trajectories. We note that we allow for $C_t < 0$, which corresponds to globally contracting maps. Given (B.4), we now define

$$
E_{s,t} = \mathbb{E}\big[\big|X_{s,t}(I_s) - \hat{X}_{s,t}(I_s)\big|^2\big],
\tag{B.5}
$$

where we recall that $X_{s,t}(x)$ satisfies $\partial_t X_{s,t}(x) = b_t(X_{s,t}(x))$ and $X_{s,s}(x) = x$. Taking the derivative with respect to $t$ of (B.5), we deduce

$$
\begin{aligned}
\partial_t E_{s,t} &= 2\mathbb{E}\big[\big(X_{s,t}(I_s) - \hat{X}_{s,t}(I_s)\big) \cdot \big(b_t(X_{s,t}(I_s)) - \partial_t \hat{X}_{s,t}(I_s)\big)\big], \\
&= 2\mathbb{E}\big[\big(X_{s,t}(I_s) - \hat{X}_{s,t}(I_s)\big) \cdot \big(b_t(\hat{X}_{s,t}(I_s)) - \partial_t \hat{X}_{s,t}(I_s)\big) \\
&\quad + 2\mathbb{E}\big[\big(X_{s,t}(I_s) - \hat{X}_{s,t}(I_s)\big) \cdot \big(b_t(X_{s,t}(I_s)) - b_t(\hat{X}_{s,t}(I_s))\big)\big], \\
&\leqslant \mathbb{E}\big[\big|X_{s,t}(I_s) - \hat{X}_{s,t}(I_s)\big|^2\big] + \mathbb{E}\big[\big|b_t(\hat{X}_{s,t}(I_s)) - \partial_t \hat{X}_{s,t}(I_s)\big|^2\big] \\
&\quad + 2\mathbb{E}\big[\big(X_{s,t}(I_s) - \hat{X}_{s,t}(I_s)\big) \cdot \big(b_t(X_{s,t}(I_s)) - b_t(\hat{X}_{s,t}(I_s))\big)\big], \\
&\equiv E_{s,t} + \delta_{s,t}^{\mathsf{LMD}} + 2\mathbb{E}\big[\big(X_{s,t}(I_s) - \hat{X}_{s,t}(I_s)\big) \cdot \big(b_t(X_{s,t}(I_s)) - b_t(\hat{X}_{s,t}(I_s))\big)\big].
\end{aligned}
\tag{B.6}
$$

Above, we defined the two-time Lagrangian distillation error,

$$
\delta_{s,t}^{\mathsf{LMD}} = \mathbb{E}\big[\big|b_t(\hat{X}_{s,t}(I_s)) - \partial_t \hat{X}_{s,t}(I_s)\big|^2\big].
\tag{B.7}
$$

By definition, the LMD loss can be expressed as $L_{\mathsf{LMD}}(\hat{X}) = \int_{[0,T]^2} w_{s,t} \delta_{s,t}^{\mathsf{LMD}} ds dt$. Using (3.5) in (B.6), we obtain the relation

$$
\partial_t E_{s,t} \leqslant (1 + 2C_t) E_{s,t} + \delta_{s,t}^{\mathsf{LMD}},
\tag{B.8}
$$

which implies that

$$
\partial_t \big(e^{-t-2\int_s^t C_u du} E_{s,t}\big) \leqslant e^{-t-2\int_s^t C_u du} \delta_{s,t}^{\mathsf{LMD}}.
\tag{B.9}
$$

Since $E_{s,s} = 0$ this implies that

$$
E_{s,t} \leqslant \int_s^t e^{(t-u)+2\int_u^t C_t dt} \delta_{s,u}^{\mathsf{LMD}} du \leqslant e^{t+2\int_s^t |C_t| dt} \int_s^t \delta_{s,u}^{\mathsf{LMD}} du.
\tag{B.10}
$$

Above, we used that $(t, u) \in [0, t]^2$ so that $(t - u) \leqslant t$. This bound shows that $E_{0,1} \leqslant e^{1+2\int_0^1 |C_t| dt} \int_0^1 \delta_{0,u}^{\mathsf{LMD}} du$, which can be written explicitly as (using $t$ instead of $u$ as dummy integration variable)

$$
\mathbb{E}\big[\big|X_{0,1}(x_0) - \hat{X}_{0,1}(x_0)\big|^2\big] \leqslant e^{1+2\int_0^1 |C_t| dt} \int_0^1 \mathbb{E}\big[\big|b_t(\hat{X}_{0,t}(x_0)) - \partial_t \hat{X}_{0,t}(x_0)\big|^2\big] dt,
\tag{B.11}
$$

Now, observe that by definition,

$$
W_2^2(\rho_1, \hat{\rho}_1) \leqslant \mathbb{E}\big[\big|X_{0,1}(x_0) - \hat{X}_{0,1}(x_0)\big|^2\big],
\tag{B.12}
$$

because the left-hand side is the infimum over all couplings and the right-hand side corresponds to a specific coupling that pairs points from the same initial condition. This completes the proof. $\qquad\square$

**Proposition 3.10** (Eulerian error bound). *Let $X_{s,t} : \mathbb{R}^d \to \mathbb{R}^d$ denote the flow map for $b$, and let $\hat{X}_{s,t} : \mathbb{R}^d \to \mathbb{R}^d$ denote an approximate flow map. Given $x_0 \sim \rho_0$, let $\hat{\rho}_1$ be the density of $\hat{X}_{0,1}(x_0)$ and let $\rho_1$ be the target density of $X_{0,1}(x_0)$. Then,*

$$
W_2^2(\rho_1, \hat{\rho}_1) \leqslant e^1 \mathcal{L}_{\mathsf{EMD}}(\hat{X}).
\tag{3.16}
$$

*Proof.* We first define the error metric

$$E_{s,t} = \mathbb{E}\left[\left|X_{s,t}(I_s) - \hat{X}_{s,t}(I_s)\right|^2\right]. \tag{B.13}$$

It then follows by direct differentiation that

$$
\begin{aligned}
\partial_s E_{s,t} &= \mathbb{E}\left[2\left(X_{s,t}(I_s) - \hat{X}_{s,t}(I_s)\right) \cdot \left(\partial_s X_{s,t}(I_s) + \dot{I}_s \cdot \nabla X_{s,t}(I_s) - \left(\partial_s \hat{X}_{s,t}(I_s) + \dot{I}_s \cdot \nabla \hat{X}_{s,t}(I_s)\right)\right)\right], \\
&= \mathbb{E}\left[2\left(X_{s,t}(I_s) - \hat{X}_{s,t}(I_s)\right) \cdot \left(\partial_s X_{s,t}(I_s) + b_s(I_s) \cdot \nabla X_{s,t}(I_s) - \left(\partial_s \hat{X}_{s,t}(I_s) + b_s(I_s) \cdot \nabla \hat{X}_{s,t}(I_s)\right)\right)\right], \\
&\geqslant -E_{s,t} - \mathbb{E}\left[\left|\partial_s X_{s,t}(I_s) + b_s(I_s) \cdot \nabla X_{s,t}(I_s) - \left(\partial_s \hat{X}_{s,t}(I_s) + b_s(I_s) \cdot \nabla \hat{X}_{s,t}(I_s)\right)\right|^2\right], \\
&= -E_{s,t} - \delta_{s,t}^{\mathsf{EMD}}.
\end{aligned}
$$

Above, we used the tower property of the conditional expectation, the Eulerian equation $\partial_s X_{s,t}(I_s) + b_s(I_s) \cdot \nabla X_{s,t}(I_s) = 0$, and defined the two-time Eulerian distillation error,

$$\delta_{s,t}^{\mathsf{EMD}} = \mathbb{E}\left[\left|\partial_s \hat{X}_{s,t}(I_s) + b_s(I_s) \cdot \nabla \hat{X}_{s,t}(I_s)\right|^2\right]. \tag{B.14}$$

This implies that

$$\partial_s \left(-e^s E_{s,t}\right) \leqslant e^s \delta_{st}^{\mathsf{EMD}}. \tag{B.15}$$

Using that $E_{t,t} = 0$ for any $t \in [0, T]$ and integrating with respect to $s$ from $s$ to $t$,

$$-e^t E_{t,t} + e^s E_{s,t} \leqslant \int_s^t e^u \delta_{u,t}^{\mathsf{EMD}} du. \tag{B.16}$$

It then follows that

$$E_{s,t} \leqslant \int_s^t e^{u-s} \delta_{u,t}^{\mathsf{EMD}} du, \tag{B.17}$$

and hence, using that $u - s \in [0, t]$ and that $\delta_{u,t}^{\mathsf{EMD}} \geqslant 0$,

$$\mathbb{E}\left[\left|X_{0,1}(x_0) - \hat{X}_{0,1}(x_0)\right|^2\right] \leqslant e^1 \int_0^1 \mathbb{E}\left[\left|\partial_s \hat{X}_{s,1}(I_s) + b_s(I_s) \cdot \nabla \hat{X}_{s,1}(I_s)\right|^2\right] ds. \tag{B.18}$$

The proof is completed upon noting that

$$W_2^2(\rho_1, \hat{\rho}_1) \leqslant \mathbb{E}\left[\left|X_{0,1}(x_0) - \hat{X}_{0,1}(x_0)\right|^2\right], \tag{B.19}$$

because the left-hand side is the infimum over all couplings and the right-hand side corresponds to a particular coupling. $\qquad\square$

**Proposition 3.11** (Flow map matching)**.** *The flow map is the global minimizer over $\hat{X}$ of the loss*

$$\mathcal{L}_{\mathsf{FMM}}(\hat{X}) = \int_{[0,1]^2} w_{s,t} \left(\mathbb{E}\left[|\partial_t \hat{X}_{s,t}(\hat{X}_{t,s}(I_t)) - \dot{I}_t|^2\right] + \mathbb{E}\left[|\hat{X}_{s,t}(\hat{X}_{t,s}(I_t)) - I_t|^2\right]\right) ds dt, \tag{3.17}$$

*subject to the boundary condition $\hat{X}_{s,s}(x) = x$ for all $x \in \mathbb{R}^d$ and for all $s \in \mathbb{R}$. In (3.17), $w_{s,t} > 0$ and $\mathbb{E}$ is taken over the coupling $(x_0, x_1) \sim \rho(x_0, x_1)$ and $z \sim \mathsf{N}(0, Id)$.*

*Proof.* We start by noticing that

$$
\begin{aligned}
&\mathbb{E}\left[|\partial_t \hat{X}_{s,t}(\hat{X}_{t,s}(I_t)) - \dot{I}_t|^2\right], \\
&= \mathbb{E}\left[|\partial_t \hat{X}_{s,t}(\hat{X}_{t,s}(I_t))|^2 - 2\dot{I}_t \cdot \partial_t \hat{X}_{s,t}(\hat{X}_{t,s}(I_t)) + |\dot{I}_t|^2\right], \\
&= \mathbb{E}\left[|\partial_t \hat{X}_{s,t}(\hat{X}_{t,s}(I_t))|^2 - 2\mathbb{E}[\dot{I}_t|I_t] \cdot \partial_t \hat{X}_{s,t}(\hat{X}_{t,s}(I_t)) + |\dot{I}_t|^2\right], \\
&= \mathbb{E}\left[|\partial_t \hat{X}_{s,t}(\hat{X}_{t,s}(I_t))|^2 - 2b_t(I_t) \cdot \partial_t \hat{X}_{s,t}(\hat{X}_{t,s}(I_t)) + |\dot{I}_t|^2\right],
\end{aligned} \tag{B.20}
$$

where we used the tower property of the conditional expectation to get the third equality and the definition of $b_t(x)$ in (3.3) to get the last. This means that the loss (3.17) can be written as

$$
\begin{aligned}
\mathcal{L}_{\mathsf{FMM}}(\hat{X}) \\
= \int_{[0,1]^2} \int_{\mathbb{R}^d} w_{s,t} \big[ |\partial_t \hat{X}_{s,t}(\hat{X}_{t,s}(x)) - b_t(x)|^2 + |\hat{X}_{s,t}(\hat{X}_{t,s}(x)) - x|^2 \big] \rho_t(x) dx ds dt \\
+ \int_{[0,1]^2} w_{s,t} \mathbb{E} \big[ |\dot{I}_t|^2 - |b_t(I_t)|^2 \big] ds dt,
\end{aligned}
\tag{B.21}
$$

where $\rho_t = \mathrm{Law}(I_t)$. The second integral does not depend on $\hat{X}$, so it does not affect the minimization of $\mathcal{L}_{\mathsf{FMM}}(\hat{X})$. Assuming that $w_{s,t} > 0$, the first integral is minimized if and only if we have

$$
\forall \ (s,t,x) \in [0,1]^2 \times \mathbb{R}^d \ : \quad \partial_t \hat{X}_{s,t}(\hat{X}_{t,s}(x)) = b_t(x) \quad \text{and} \quad \hat{X}_{s,t}(\hat{X}_{t,s}(x)) = x.
\tag{B.22}
$$

From the second of these equations it follows that: (i) $\hat{X}_{s,s}(x) = x$, and (ii) if we evaluate the first equation at $y = \hat{X}_{t,s}(x)$, this equation reduces to

$$
\forall \ (s,t,y) \in [0,1]^2 \times \mathbb{R}^d \ : \quad \partial_t \hat{X}_{s,t}(y) = b_t(\hat{X}_{s,t}(y))
\tag{B.23}
$$

which recovers (3.7). $\qquad\square$

**Proposition 3.12** (Eulerian estimation)**.** *The flow map $X_{s,t}$ is a critical point of the loss*

$$
\mathcal{L}_{\mathsf{EE}}(\hat{X}) = \int_{[0,1]^2} w_{s,t} \mathbb{E} \big[ |\partial_s \hat{X}_{s,t}(I_s) - \mathsf{stopgrad}(\dot{I}_s \cdot \nabla \hat{X}_{s,t}(I_s))|^2 \big] ds dt,
\tag{3.22}
$$

*subject to the boundary condition $\hat{X}_{s,s}(x) = x$ for all $x \in \mathbb{R}^d$ and for all $s \in \mathbb{R}$. In (3.22), $w_{s,t} > 0$ and $\mathbb{E}$ is taken over the coupling $(x_0, x_1) \sim \rho(x_0, x_1)$, $z \sim \mathsf{N}(0, Id)$. The operator $\mathsf{stopgrad}(\cdot)$ indicates that the argument is ignored when computing the gradient.*

*Proof.* The functional gradient of (3.22) with respect to $\hat{X}_{s,t}$ is given by

$$
\begin{aligned}
&- \partial_s \big( w_{s,t} \big[ \partial_s \hat{X}_{s,t}(x) + \mathbb{E} \big[ \dot{I}_s \cdot \nabla \hat{X}_{s,t}(I_s) \mid I_s = x \big] \big] \rho_s(x) \big) \\
&= - \partial_s \big( w_{s,t} \big[ \partial_s \hat{X}_{s,t}(x) + b_s(x) \cdot \nabla \hat{X}_{s,t}(x) \big] \rho_s(x) \big).
\end{aligned}
\tag{B.24}
$$

The flow map zeroes this quantity since it solves the Eulerian equation (3.10) that appears in it, and hence is a critical point of the objective. $\qquad\square$

**Lemma 3.13** (Progressive flow map matching)**.** *Let $\hat{X}$ be a two-time flow map. Given $K \in \mathbb{N}$, let $t_k = s + \frac{k-1}{K-1}(t-s)$ for $k = 1, \ldots, K$. Then the unique minimizer over $\check{X}$ of the objective*

$$
\mathcal{L}_{\mathsf{PFMM}}(\check{X}) = \int_{[0,1]^2} w_{s,t} \mathbb{E} \Big[ \big| \check{X}_{s,t}(I_s) - \big( \hat{X}_{t_{K-1},t_K} \circ \cdots \circ \hat{X}_{t_1,t_2} \big)(I_s) \big|^2 \Big] ds dt,
\tag{3.23}
$$

*produces the same output in one step as the $K$-step iterated map $\hat{X}$. Here $w_{s,t} > 0$, and $\mathbb{E}$ is taken over the coupling $(x_0, x_1) \sim \rho(x_0, x_1)$ and $z \sim \mathsf{N}(0, Id)$.*

*Proof.* Equation (3.23) is a PINN loss whose unique minimizer satisfies

$$
\forall \ (s,t,x) \in [0,1]^2 \times \mathbb{R}^d \ : \quad \check{X}_{s,t}(x) = \big( \hat{X}_{t_{K-1},t_K} \circ \cdots \circ \hat{X}_{t_1,t_2} \big)(x),
\tag{B.25}
$$

which establishes the claim. $\qquad\square$

# C   Relation to existing consistency and distillation techniques

In this section, we recast consistency models and several distillation techniques in the language of our two-time flow map $X_{s,t}$ to clarify their relation with our work.

### C.1 Flow maps and denoisers

Since $\mathrm{Law}(X_{t,s}(I_t)) = \mathrm{Law}(I_s)$, it is tempting to replace $X_{t,s}(I_t)$ by $I_s$ in the loss (3.17) and use instead

$$\mathcal{L}_{\mathrm{denoise}}[\hat{X}] = \int_{[0,1]^2} w_{s,t} \mathbb{E}\big[|\partial_t \hat{X}_{s,t}(I_s) - \dot{I}_t|^2\big] ds dt, \tag{C.1}$$

minimized over all $\hat{X}$ such that $\hat{X}_{s,s}(x) = x$. However, the minimizer of this objective is *not* the flow map $X_{s,t}$, but rather the denoiser

$$X_{s,t}^{\mathrm{denoise}}(x) = \mathbb{E}[I_t | I_s = x]. \tag{C.2}$$

This can be seen by noticing that the minimizer of (C.1) is the same as the minimizer of

$$\begin{aligned}
\mathcal{L}'_{\mathrm{denoise}}[\hat{X}] &= \int_{[0,1]^2} w_{s,t} \mathbb{E}\big[\big|\partial_t \hat{X}_{s,t}(I_s) - \mathbb{E}[\dot{I}_t | I_s]\big|^2\big] ds dt, \\
&= \int_{[0,1]^2} \int_{\mathbb{R}^d} w_{s,t}\Big[\big|\partial_t \hat{X}_{s,t}(x) - \mathbb{E}[\dot{I}_t | I_s = x]\big|^2\Big] \rho_s(x) dx ds dt,
\end{aligned} \tag{C.3}$$

which follows from an argument similar to the one used in the proof of Proposition 3.11. The minimizer of (C.3) satifies

$$\partial_t \hat{X}_{s,t}(x) = \mathbb{E}[\dot{I}_t | I_s = x] = \partial_t \mathbb{E}[I_t | I_s = x], \tag{C.4}$$

which implies (C.2) by the boundary condition $\hat{X}_{s,s}(x) = x$. The denoiser (C.2) may be useful, but it is not a consistent generative model. For instance, if $x_0 \sim \rho_0$ and $x_1 \sim \rho_1$ are independent in the definition of $I_t$, since $I_0 = x_0$ and $I_1 = x_1$ by construction, for $s = 0$ and $t = 1$ we have

$$X_{0,1}^{\mathrm{denoise}}(x) = \mathbb{E}[x_1] \tag{C.5}$$

i.e. the one-step denoiser only recovers the mean of the target density $\rho_1$.

### C.2 Relation to consistency models

**Noising process.** Following the recommendations in Karras et al. (2022) (which are followed by both Song et al. (2023) and Song and Dhariwal (2023)), we consider the variance-exploding process[4]

$$\tilde{x}_t = a + tz, \quad t \in [0, t_{\max}], \tag{C.6}$$

where $a \sim \rho_1$ (data from the target density) and $z \sim \mathsf{N}(0, I)$. In practice, practitioners often set $t_{\max} = 80$. In this section, because we follow the score-based diffusion convention, we set time so that $t = 0$ recovers $\rho_1$ and so that a Gaussian is recovered as $t \to \infty$. The corresponding probability flow ODE is given by

$$\dot{\tilde{x}}_t = -t \nabla \log \rho_t(\tilde{x}_t), \qquad \tilde{x}_{t=0} = a \sim \rho_1 \tag{C.7}$$

where $\rho_t(x) = \mathrm{Law}(\tilde{x}_t)$. In practice, (C.7) is solved backwards in time from some terminal condition $\tilde{x}_{t_{\max}}$. To make contact with our formulation where time goes forward, we define $x_t = \tilde{x}_{t_{\max}-t}$, leading to

$$\dot{x}_t = (t_{\max} - t) \nabla \log \rho_{t_{\max}-t}(x_t), \qquad x_{t=0} \sim \mathsf{N}(x_0, t_{\max}^2 I). \tag{C.8}$$

To make touch with our flow map notation, we then define

$$\partial_t X_{s,t}(x) = (t_{\max} - t) \nabla \log \rho_{t_{\max}-t}(X_{s,t}(x)), \qquad X_{s,s}(x) = x. \tag{C.9}$$

---

[4]Oftentimes $t = 0$ is set to $t = t_{\min} > 0$ for numerical stability, choosing e.g. $t_{\min} = 2 \times 10^{-3}$.

**Consistency function.** By definition (Song et al., 2023), the consistency function $f_t : \mathbb{R}^d \to \mathbb{R}^d$ is such that

$$f_t(\tilde{x}_t) = a, \tag{C.10}$$

where $\tilde{x}_t$ denotes the solution of (C.7) and $a \sim \rho_1$. To make a connection with our flow map formulation, let us consider (C.10) from the perspective of $x_t$,

$$f_t(x_{t_{\max}-t}) = x_{t_{\max}}, \tag{C.11}$$

which is to say that

$$f_t(x) = X_{t_{\max}-t,t_{\max}}(x). \tag{C.12}$$

Note that only one time is varied here, i.e. $f_t(x)$, cannot be iterated upon: by its definition (C.10), it always maps the observation $\tilde{x}_t$ onto a sample $a \sim \rho_1$.

**Discrete-time loss function for distillation.** In practice, consistency models are typically trained in discrete-time, by discretizing $[t_{\min}, t_{\max}]$ into a set of $N$ points $t_{\min} = t_1 < t_2 < \ldots < t_N = t_{\max}$. According to Karras et al. (2022), these points are chosen as

$$t_i = \left( t_{\min}^{1/\eta} + \frac{i-1}{N-1} \left( t_{\max}^{1/\eta} - t_{\min}^{1/\eta} \right) \right)^{\eta}, \tag{C.13}$$

with $\eta = 7$. Assuming that we have at our disposal a pre-trained estimate $s_t(x)$ of the score $\nabla \log \rho_t(x)$, the *distillation loss* for the consistency function $f_t(x)$ is then given by

$$\begin{aligned}
\mathcal{L}_{\mathsf{CD}}^N(\hat{f}) &= \sum_{i=1}^{N-1} \mathbb{E}\left[ \left| \hat{f}_{t_{i+1}}(\tilde{x}_{t_{i+1}}) - \hat{f}_{t_i}(\hat{x}_{t_i}) \right|^2 \right], \\
\tilde{x}_{t_{i+1}} &= a + t_{i+1}z \\
\hat{x}_{t_i} &= \tilde{x}_{t_{i+1}} - (t_i - t_{i+1}) t_{i+1} s_{t_{i+1}}(x_{t_{i+1}}),
\end{aligned} \tag{C.14}$$

where $\mathbb{E}$ is taken over the data $a \sim \rho_1$ and $z \sim \mathsf{N}(0, I)$. The term $\hat{x}_{t_i}$ is an approximation of $\tilde{x}_{t_i}$ computed by taking a single step of (C.7) with the approximate score model $s_t(x)$. In practice, the square loss in (C.14) can be replaced by an arbitrary metric $d : \mathbb{R}^d \to \mathbb{R}^d \to \mathbb{R}_{\geqslant 0}$, such as a learned metric like LPIPS or the Huber loss.

**Continuous-time limit.** In continuous-time, it is easy to see via Taylor expansion that the consistency loss reduces to

$$\mathcal{L}_{\mathsf{CD}}^\infty(\hat{f}) = \lim_{N \to \infty} N \mathcal{L}_{\mathsf{CD}}^N(\hat{f}) = \int_{t_{\min}}^{t_{\max}} \int_{\mathbb{R}^d} w_t^2 \left| \partial_t f_t(x) - t s_t(x) \cdot \nabla f_t(x) \right|^2 \rho_t(x) dx dt, \tag{C.15}$$

where $w_t = \eta(t_{\max}^{1/\eta} - t_{\min}^{1/\eta}) t^{1-1/\eta}$ is a weight factor arising from the nonuniform time-grid. This is a particular case of our Eulerian distillation loss (3.13) applied to the variance-exploding setting (C.6) with the identification (C.12).

**Estimation vs distillation of the consistency model.** If we approximate the exact

$$\nabla \log \rho_t(x) = -\mathbb{E}\left[ \frac{\tilde{x}_t - a}{t^2} \middle| \tilde{x}_t = x \right], \tag{C.16}$$

by a single-point estimator

$$\nabla \log \rho_t(x) \approx \frac{a - \tilde{x}_t}{t^2}, \tag{C.17}$$

we may use the expression

$$\hat{x}_{t_i} \approx a + t_i z, \tag{C.18}$$

in (C.14) to obtain the *estimation* loss,

$$\mathcal{L}_{\mathsf{CT}}^N(\hat{f}) = \sum_{i=1}^{N-1} \mathbb{E}\Big[\big|\hat{f}_{t_{i+1}}(\tilde{x}_{t_{i+1}}) - \hat{f}_{t_i}^-(\tilde{x}_{t_i})\big|^2\Big], \tag{C.19}$$
$$\tilde{x}_{t_i} = a + t_i z.$$

This expression does not require a previously-trained score model. Notice, however, that (C.19) must be used with a stopgrad on $\hat{f}_{t_i}^-(\tilde{x}_{t_i})$ so that the gradient is taken over only the first $\hat{f}_{t_{i+1}}(\tilde{x}_{t_{i+1}})$. This is because (C.14) and (C.19) are different objectives with different minimizers, even at leading order after expansion in $1/N$, for the same reason that (3.13) differs from (3.18). To see this, observe that to leading order,

$$\hat{f}_{t_i}^-(\tilde{x}_{t_i}) = \hat{f}_{t_{i+1}}^-(\tilde{x}_{t_{i+1}}) + \big(\partial_t \hat{f}_{t_{i+1}}^-(\tilde{x}_{t_{i+1}}) + z \cdot \nabla f_{t_{i+1}}^-(\tilde{x}_{t_{i+1}})\big)(t_i - t_{i+1}) + O\big((t_i - t_{i+1})^2\big), \tag{C.20}$$

which gives the continuous-time limit

$$\mathcal{L}_{\mathsf{CT}}^\infty(\hat{f}) = \lim_{N \to \infty} \mathcal{L}_{\mathsf{CD}}^N(\hat{f}) = \int_{t_{\min}}^{t_{\max}} \int_{\mathbb{R}^d} w_t \big|\partial_t f_t(x) + z \cdot \nabla f_t^-(x)\big|^2 \rho_t(x) dx dt. \tag{C.21}$$

Observing that $z = \partial_t \tilde{x}_t$ shows that (C.21) recovers the Eulerian estimator described in Section 3.6, which does not lead to a gradient descent iteration.

## C.3 Relation to neural operators

In our notation, neural operator approaches for fast sampling of diffusion models (Zheng et al., 2023) also estimate the flow map $X_{0,t}$ via the loss

$$\mathcal{L}_{\mathsf{FNO}}(\hat{X}) = \int_0^1 \int_{\mathbb{R}^d} \big|\hat{X}_{0,t}(x) - X_{0,t}(x)\big|^2 \rho_0(x) dx dt, \tag{C.22}$$

where $\hat{X}_{0,t}$ is parameterized by a Fourier Neural Operator and where $X_{0,t}$ is the flow map *obtained by simulating the probability flow ODE associated with a pre-trained (or given) $b_t(x)$*. To avoid simulation at learning time, they pre-generate a dataset of trajectories, giving access to $X_{0,t}(x)$ for many initial conditions $x \sim \rho_0$. Much of the work focuses on the architecture of the FNO itself, which is combined with a U-Net.

## C.4 Relation to progressive distillation

Progressive distillation (Salimans and Ho, 2022) takes a DDIM sampler (Song et al., 2022) and trains a new model to approximate two steps of the old sampler with one step of the new model. This process is iterated repeatedly to successively halve the number of steps required. In our notation, this corresponds to minimizing

$$\mathcal{L}_{\mathsf{PD}}^{\Delta t}(\hat{X}) = \int_0^{1-2\Delta t} \int_{\mathbb{R}^d} \big|\hat{X}_{t,t+2\Delta t}(x) - \big(X_{t+\Delta t,t+2\Delta t} \circ X_{t,t+\Delta t}\big)(x)\big|^2 \rho_t(x) dx dt \tag{C.23}$$

where $X$ is a pre-trained map from the previous iteration. This is then iterated upon, and $\Delta t$ is increased, until what is left is a few-step model.

# D Additional Experimental Details

## D.1 2D checkerboard

Here, we provide further discussion and analysis of our results for generative modeling on the 2D checkerboard distribution (Figure 3). Our KL-divergence estimates clearly highlight that there is a hierarchy of performance. Of particular interest is the large discrepancy in performance between the Eulerian and Lagrangian distillation techniques.

|              | $\mathsf{KL}(\rho_1\|\hat{\rho}_1)$ | $W_2^2(\rho_1,\hat{\rho}_1)$ | $W_2^2(\hat{\rho}_1^b,\hat{\rho}_1)$ | $L_2$ error |
|--------------|-------------------|-----------------|------------------|-------------|
| SI           | 0.020             | 0.026           | 0.0              | 0.000       |
| LMD          | 0.043             | 0.059           | 0.032            | 0.085       |
| EMD          | 0.079             | 0.029           | 0.010            | 0.011       |
| FMM, $N=1$   | 0.104             | 0.021           | –                | 0.026       |
| FMM, $N=4$   | 0.045             | 0.014           | –                | 0.024       |
| PFMM, $N=1$  | 0.043             | 0.014           | –                | 0.023       |

**Table 3:** Comparison of $\mathsf{KL}(\rho\|\hat{\rho})$ and $W_2^2(\rho,\hat{\rho})$, where $\hat{\rho}$ is the pushforward density from the maps $\hat{X}_{0,1}(x_0)$ for the methods listed above. Additionally included is a comparison of $L_2$ expected error of the distillation methods against their teacher $\hat{X}_{0,1}^{\mathsf{SI}}$ given as $\mathbb{E}[|\hat{X}_{0,1}^{\mathsf{SI}}(x) - \hat{X}_{0,1}(x)|^2]$. Intriguingly, LMD performs better in being distributionally correct, as measured by the KL-divergence, but worse in preserving the coupling of the teacher model. The roles are flipped for EMD. This may highlight KL as a more informative metric in our case, as our aims are to sample correctly in distribution. See Figure 6 for a visualization.

As noted in Figure 3 and Table 3, LMD substantially outperforms its Eulerian counterpart in terms of minimizing the KL-divergence between the target checkerboard density $\rho_1$ and model density $\hat{\rho}_1 = \hat{X}_{0,1}\sharp\rho_0$. Interestingly, while LMD is more correct in distribution, EMD better preserves the original coupling $(x_0, \hat{X}_{0,1}^{\mathsf{SI}}(x_0))$ of the teacher model $\hat{X}_{0,1}^{\mathsf{SI}}$, as measured by the $W_2^2$ distance and the expected $L_2$ reconstruction error, defined as $\mathbb{E}[|\hat{X}_{0,1}^{\mathsf{SI}}(x) - \hat{X}_{0,1}(x)|^2]$. Where this coupling is significantly *not* preserved is visualized in Figure 6. For each model, we color code points for which $|\hat{X}_{0,1}^{\mathsf{SI}}(x) - \hat{X}_{0,1}(x)|^2 > 1.0$, highlighting where the student map differed from the teacher. We notice that the LMD map pushes initial conditions to an opposing checker edge (purple) than where those initial conditions are pushed by the interpolant (blue). This is much less common for the EMD map, but its performance is overall worse in matching the target distribution.

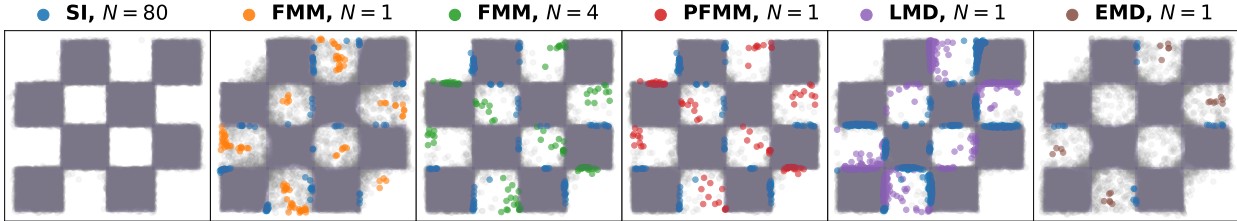

**Figure 6:** Visualization of the difference in assignment of the maps $\hat{X}_{0,1}(x_0)$ for the various models as compared to the teacher/ground truth model $\hat{X}_{0,1}(x_0)$ for the same initial conditions from the base distribution $x_0$. Points that lie in the region $|\hat{X}_{0,1}^{\mathsf{SI}}(x_0) - \hat{X}_{0,1}(x_0)|^2 > 1.0$ are colored as compared to the blue points, which represent where the stochastic interpolant teacher mapped the same red initial conditions. This gives us an intuition for how well each method precisely maintains the coupling $(x_0, \hat{X}_{0,1}^{\mathsf{SI}}(x_0))$ from the teacher. Note that we are treating $X_{0,1}^{\mathsf{SI}}$ as the ground truth map here, as it is close to the exact map. The models based on FMM either don't have a teacher or have FMM, $N=4$ as a teacher, but all should have the same coupling at the minimizer.

## D.2   Image experiments

Here we include more experimental details for reproducing the results provided in Section 4. We use the U-Net from the diffusion OpenAI paper (Dhariwal and Nichol, 2021) with code given at https://github.com/openai/guided-diffusion. We use the same architecture for both CIFAR10 and ImageNet-$32 \times 32$ experiments. The architecture is also the same for training a velocity field and for training a flow map, barring the augmentation of the time-step embedding in the U-Net to handle two times $(s,t)$ instead of one. Details of the training conditions are presented in Table 4.

|  | CIFAR-10 | ImageNet 32×32 |
|---|---|---|
| Dimension | 32×32 | 32×32 |
| # Training point | $5 \times 10^4$ | 1,281,167 |
| Batch Size | 256 | 256 |
| Training Steps (Lagrangian distillation) | $1.5 \times 10^5$ | $2.5 \times 10^5$ |
| Training Steps (Eulerian distillation) | $1.2 \times 10^5$ | $2.5 \times 10^5$ |
| Training Steps (Flow map matching) | N/A | $1 \times 10^5$ |
| Training Steps (Progressive flow map matching) | $1.3 \times 10^5$ | N/A |
| U-Net channel dims | 256 | 256 |
| Learning Rate (LR) | 0.0001 | 0.0001 |
| LR decay (every 1k epochs) | 0.992 | 0.992 |
| U-Net dim mult | [1,2,2,2] | [1,2,2,2] |
| Learned time embedding | Yes | Yes |
| # GPUs | 4 | 4 |

**Table 4:** Hyperparameters and architecture for image datasets.

