# OpenReview forum: "Flow map matching with stochastic interpolants: A mathematical framework for consistency models"
_TMLR — Accepted by TMLR_

### Review · Reviewer_bE2k · 2024-09-01

**Summary Of Contributions:**

Authors propose using two types of losses (Lagragian map distillation-based loss and Eulerian map distillation-based loss) to obtain a flow map, which is an image generation function.

The formulation of these two distillation losses are motivated from an autonomous system ODE and conditions of density functions of incompressible flow.

The main gap authors want to bridge is the necessity of multiple steps of generation from extant methods.

Authors presented experimental results to generate synthetic images similar to those of CIFAR10 and ImageNet, and showed Frechet Inception Distance metrics to show the generated images are of better quality compared to those by DDPM and BatchOT.

**Audience:**

Yes

**Claims And Evidence:**

No

**Requested Changes:**

Questions:

- How is the _velocity field_ $b_t$ pre-trained? In the context of image generation, for instance based on CIFAR-10 and ImageNet datasets, how is this $b_t$ pre-trained and what physical meaning does it have?

- In section 3.4, how is $\dot{I}_t$ computed? Even taking a step back, please explain clearly the relationship between $I_t$, $\rho_t$ and $b_t$. How are they pre-computed? How are they prepared in the numerical experiments for PFMM and FMM methods, respectively?

- What is "_identification $t$_" above proposition 3.11?

- In order to make method comparison fair, also to highlight the main motivation of this work, does it make sense to benchmark with other methods that also claim the benefit of few-step generations? See [1-8] and references in the Related Work section in [1]. I am not asking authors to make comparison to all those methods, while some other methods which aim to address the computationally expensive multi-step simulation problems should be proper baselines.

- The benefits or drawbacks of the proposed methods are not clearly explained in the experimental section. What is the "_Baseline_" in Table 1. Table 1 shows that in terms of the FID metric, proposed methods underperforms against the "_Baseline_".

Authors should give more careful thoughts distinguishing the advantages and disadvantages of each proposed method, and give a characterisation on when to use which. Roughly talking about their computational time difference is insufficient. It leaves the impression that methods are simply stacked up without being well understood.


Minor comments:

- Define notations used in the manuscript clearly, such as $X_{0,1}\sharp\rho_0$, $\perp$, $\mathrm{Law}(I_t)$.

- Define clearly what $\hat{X}$ and $\check{X}$ are.


*************************************
[1] Clément Chadebec, Onur Tasar, Eyal Benaroche, Benjamin Aubin, Flash Diffusion: Accelerating Any Conditional Diffusion Model for Few Steps Image Generation, 2024, arXiv:2406.02347v2

[2] Yin et al., One-step Diffusion with Distribution Matching Distillation, arXiv preprint arXiv:2311.18828, 2023.

[3] Shanchuan Lin, Anran Wang, and Xiao Yang. Sdxl-lightning: Progressive adversarial diffusion distillation. arXiv preprint arXiv:2402.13929, 2024.

[4] Xingchao Liu, Chengyue Gong, et al. Flow straight and fast: Learning to generate and transfer data with rectified flow. In The Eleventh International Conference on Learning Representations, 2022

[5] Xingchao Liu, Xiwen Zhang, Jianzhu Ma, Jian Peng, et al. Instaflow: One step is enough for highquality diffusion-based text-to-image generation. In The Twelfth International Conference on Learning Representations, 2023.

[6] Dongjun Kim, Chieh-Hsin Lai, Wei-Hsiang Liao, Naoki Murata, Yuhta Takida, Toshimitsu Uesaka, Yutong He, Yuki Mitsufuji, and Stefano Ermon. Consistency trajectory models: Learning probability flow ode trajectory of diffusion. In The Twelfth International Conference on Learning Representations, 2023.

[7] Simian Luo, Yiqin Tan, Suraj Patil, Daniel Gu, Patrick von Platen, Apolinário Passos, Longbo Huang, Jian Li, and Hang Zhao. Lcm-lora: A universal stable-diffusion acceleration module. arXiv preprint, arXiv:2311.05556, 2023.

[8] Axel Sauer, Frederic Boesel, Tim Dockhorn, Andreas Blattmann, Patrick Esser, and Robin Rombach. Fast high-resolution image synthesis with latent adversarial diffusion distillation. arXiv preprint, arXiv:2403.12015, 2024.

**Strengths And Weaknesses:**

See requested changes section for specific questions.

---

> ### Author Response · Authors · 2025-03-11
> **Response to Reviewer bE2k**
>
> We thank the reviewer for their questions and comments, and for pointing out these additional references.
>
> - *Pretraining $b_t$*: In the revision, we have better explained how this velocity field is obtained within the context of flow matching with stochastic interpolants and score-based diffusion models. To this effect, we have modified the discussion in Sec. 3.1 and have also added a new Appendix A that summarizes the frameworks of stochastic interpolants and score-based diffusion models.
>
> - *Relation between $I_t$, $b_t$ and $\rho_t$*: The newly-added Appendix A should answer these questions in a self-contained fashion. We thank the reviewer for suggesting that we clarify these points.
>
> - *Identification $t$*: We meant that, by changing time according to $t \mapsto -\log t$, an explicit mapping between stochastic interpolants and score-based diffusion models can be constructed. The relation between score-based diffusion models and stochastic interpolants is now discussd in detail in Appendix A.4.
>
> - *Baselines and comparison with Refs. [1-8]:* In principle, we agree that a systematic comparison between existing methods and our own would be a useful contribution for the community. In practice, many of the cited works perform experiments on larger-scale GPU clusters than we have access to, which makes a fair comparison on realistic datasets challenging. Instead, we have chosen to mathematically formalize the concept of flow map estimation -- which, to our knowledge, includes all known recent consistency-type models as a special case -- and compare the resulting training approaches on an equal footing. It is our firm belief that *any* method we introduced in this paper can be made to work with sufficient scale, data, and computational resources. Despite this, we show that there are distinct tradeoffs between the different approaches, as we describe more fully in the following bullet point. We have shown these tradeoffs at moderate scale ranging from two-dimensional datasets to image datasets such as CIFAR-10 and ImageNet 32x32, but we expect them to persist across distinct data modalities and at larger scale. Investigating the large-scale properties of these models is an interesting direction that we intend to pursue in future work if we are able to attain the required compute.
>
> - *Benefits, drawbacks, and advantages:* We thank the reviewer for suggesting that we clarify the relative strengths of each of the introduced methods, but we respectfully disagree that the methods are not well understood. Broadly, the methods we introduce split into two classes: those introduced for distillation of a pre-trained field, and those introduced to train from scratch. For distillation, we compare the Lagrangian loss to the Eulerian loss; the Lagrangian loss is new, while the Eulerian loss recovers consistency distillation in the continuous-time limit. On a range of datasets, we find that the Lagrangian loss outperforms its Eulerian counterpart, suggesting that it is superior in the distillation setting. This has important implications for training of consistency models, as it can significantly improve performance at no additional computational expense. In Table 1, for these distillation-based techniques, the "Baseline" refers to a flow matching model integrated with many timesteps. Therefore, we expect the student model to perform on-par or slightly worse, as it cannot exceed the performance of the teacher and it generates samples at ~20x less cost. This is precisely what the results in the table show. On the other hand, many researchers are interested in distillation-free approaches, so that the performance of the student is not limited by that of the teacher. We introduce a novel training loss (and an associated progressive variant) for this setting based on the Lagrangian perspective, which we show can be used to train models directly, but which comes at a higher computational expense than the "pre-train and distill" paradigm. Moreover, we show that the same kind of training loss cannot be introduced for the Eulerian approach.
>
> - *Notations*: In the revision, we have better explained (or in some cases, eliminated) notation such as $\text{Law}(x_t)$, $X_{0,t} \sharp \rho_0$, etc. We hope that this clarifies the reviewers questions.

---

### Review · Reviewer_ZNnM · 2024-10-02

**Summary Of Contributions:**

The paper tackles the issue of evaluating generative models defined via differential equations, such as diffusion and flow matching models, sampling from which usually relies on expensive numerical integration.
To solve this problem, the authors propose to learn what they call the *two-time flow map* of the model, i.e. the integrator of the dynamics that define the generative model. In other words, instead of handling the model dynamics and integrating them directly, they propose learning a function that takes an integration start time $s$, end time $t$ and the state at the start time $x_s$ and outputs the state $x_t$ at time $t$.
They propose four possible objectives for learning the flow map based on different formulations of the problem depending on the information available to the practitioner. They benchmark their solution on learning a 2-dimensional toy distribution and an image generation problem on CIFAR-10 and ImageNet 32x32.

**Audience:**

Yes

**Claims And Evidence:**

Yes

**Requested Changes:**

Necessary:
 - Please address my concerns in the weaknesses section.
 - Bottom of page 4: "by providing access to alternative samples from" - I believe it should be "by providing an alternative to access samples from".
 - Please define what T-FID is.
 - Please increase the legend size in Fig 4/B; it should be easily readable at 100% zoom.

**Strengths And Weaknesses:**

## Strengths
Overall, the paper is nicely written and reasonably easy to follow, even for someone who is not an expert in dynamics-based generative modelling like me. The figures in the text are also very nice.

The results showing the existence and uniqueness of the solutions to their proposed objectives follow in a reasonably straightforward fashion from previous results, but they are important to establish nonetheless. I had a cursory check of the proofs, and apart from two inequalities (see below), I believe they are correct.

## Weaknesses
I don't think the paper suffers from any major weakness that should prevent it from appearing in TMLR.

However, there are a few points that could be improved to strengthen the paper:
 - Related works: The authors' proposed approach seems very similar to the work of [1] and [2], but these are not cited. Could the authors explain how their method is different from these? At a glance, it seems that the authors have more theoretical results/their framework is slightly more general, whereas these papers do more experiments.
 - I appreciated the error bounds in Section 3.3, but how tight these are is unclear. Are there comparable bounds or bounds that are special cases of the ones in the paper present in the literature? Or are these the first such error bounds to appear? In any case, the authors should note this.
 - The four objectives proposed by the authors are always compared as equals in the experiments. However, it is unclear whether this means we should always prefer the LMD formulation (the best-performing one) or if other potential factors should guide this decision. It would be great if the authors could include a paragraph or two discussing when to apply which objective or present the necessary assumptions and design decisions in a table.
 - Why do the inequalities in A.11 and between A.18 and A.19 hold?

## References:
 - [1] Li, L., & He, J. (2024). Bidirectional Consistency Models. arXiv preprint arXiv:2403.18035.
 - [2] Kim, B., Kim, J., Kim, J., & Ye, J. C. (2024). Generalized Consistency Trajectory Models for Image Manipulation. arXiv preprint arXiv:2403.12510.

---

> ### Author Response · Authors · 2025-03-11
> **Response to Reviewer ZNnM**
>
> We thank the reviewer for their questions and comments, and for pointing out these additional relevant references.
>
> - *Related work*: We have added a discussion about the two suggested references to better contextualize how these methods differ from our own. Reference [1] introduces a framework for learning the two-time flow map that can map either forwards or backwards in time, but only considers the diffusion model setting, and learns using techniques inspired by consistency models. Reference [2] learns a two-time flow map in the stochastic interpolant framework, but does not learn to map backwards in time, and similarly leverages training inspired by consistency models. The framework that we propose is more general than either of these: it can be used for distillation or direct training, can leverage new Lagrangian losses or consistency-like Eulerian losses in both cases, can be used for bidirectional or forward-only models, and can be used with diffusions or interpolants. In this sense, it extends consistency models along two independent axes, while each of the above methods extends along a single axis; as such, our approach includes both as a special case.
>
> - *Error bounds:* The error bounds that we derive are tight, in that there are easy-to-construct systems that saturate the bounds, though we expect that in practice they will often be pessimistic for real-world generative models. To the best of our knowledge, this is the first time that such bounds have been derived for consistency models, and they therefore provide some of the first theoretical guarantees for sampling with consistency models.
>
> - *Various objectives*: Our experiments suggest that the Lagrangian loss should be preferred over the more standard Eulerian loss used to train consistency models for distillation of a pre-trained velocity field. However, both losses are inapplicable when we would like to train the flow map directly without pre-training a probability flow ODE. In this case, the FMM loss is the only one that can be used in a principled way. Our results show that the FMM loss gives accurate results, either alone or in combination with our PFMM loss.
>
> - *Inequalities*: These simply use Young's inequality, namely that $\pm 2a\cdot b \le |a|^2+|b|^2$.
>
> - *Misprints:* We have fixed the misprints pointed out by the reviewers and have clarified the jargon used. We have also improved the resolution of our figures, and we thank the reviewer for the suggestion.
>
> - *Minor changes:* We have added a few sentences clearly defining the T-FID, which is the FID computed over a dataset generated by the teacher rather than the training dataset. We have also also increased the legend size in Figure 4B.

---

> > ### Comment · Reviewer_ZNnM · 2025-03-17
> >
> > I thank the authors for addressing my concerns in their rebuttal. I am happy to recommend acceptance.

---

### Review · Reviewer_NFyf · 2025-01-31

**Summary Of Contributions:**

This paper proposes Flow Map Matching (FMM) for one or few-step generation along neural ODEs. The authors show that the flow map, a function which translates points between two arbitrary time-points of neural ODEs, satisfy a certain Lagrangian equation (e.g., Eq. (3.5)). Inspired by this observation, the authors propose optimizing a neural net to satisfy the Lagrangian equation to learn the flow map. The authors numerically show that FMM is capable of learning two-dimensional distributions or 32x32 resolution images.

**Audience:**

Yes

**Broader Impact Concerns:**

None.

**Claims And Evidence:**

No

**Requested Changes:**

Addressing both weaknesses are critial to securing my recommendation for acceptance. Please provide clarifications or additional proofs to address [W1], and provide further experimental results to address [W2].

**Strengths And Weaknesses:**

**Strengths**
- [S1] To the best of my knowledge, this is the first work to leverage the Lagrangian equation to learn few-step generators along neural-ODEs.

**Weaknesses**
- [W1] Theoretical concerns. There is no proof that the flow map is the only map which satisfies the Lagrangian equation. Specifically, the authors only show that flow map satisfies the Lagrangian equation. I was unable to find a proof that a map which satisfies the Lagrangian equation is the flow map. Hence, it is difficult to see whether training a neural net to satisfy the Lagrangian equation will truly yield one-step or few-step generators along neural ODEs.
- [W2] Weak experiments. I do not find the experimental results very convincing. Indeed, in Figure 3, FMM with N=1 is highly inferior to SI, which enhances my concern in [W1] - that training a neural net to satisfy the Lagrangian equation may not yield one-step or few-step generators along neural ODEs.

---

> ### Author Response · Authors · 2025-03-11
> **Response to reviewer NFyf**
>
> We thank the reviewer for their questions and comments, which we believe have improved the clarity and results of the manuscript.
>
> [W1] *Theoretical concerns*: The flow map is the only solution to the Lagrangian equation that satisfies the boundary conditions $X_{s,s}(x) = x$ for all $s\in[0,T]$. In practice, we enforce this constraint by choice of the neural network architecture. This statement follows from standard theorems on ODEs with drifts that satisfy a one-sided Lipschitz condition (our Assumption 3.1); for further information, please see e.g. Chapter 5 of [1]. We have modified the text to emphasize this point more clearly.
>
> [W2] *Weak experiments*: The experiments reported in Fig. 3 on the checkerboard distribution were misleading, and we thank the reviewer for drawing our attention to this fact. We had intentionally benchmarked all considered methods with small networks and short training runs to better emphasize potential differences between the approaches. We have now repeated these experiments with a larger network (a residual network with 6 layers of width 512) and longer training runs. The new results show that all methods -- except the one based on the Eulerian loss (EMD) -- perform very well, with no convergence issues. Similarly, we have added a style transfer experiment on CIFAR-10 that highlights how we can leverage our framework to efficiently solve novel downstream problems.
>
> [1] Hartman, P. (2002). Ordinary differential equations (Classics in Applied Mathematics, Series No. 38). Society for Industrial and Applied Mathematics.

---

### Author Response · Authors · 2025-02-14
**Assembling review response**

Dear reviewers,

Thanks kindly for all of your feedback! We wanted to write to let you know we are assembling our response, and will send it over to you in the coming days.

Best,
The Authors

---

### Author Response · Authors · 2025-03-11
**General Response to Reviewers**

We thank the reviewers for their careful reading of our paper and for their constructive comments and questions. In the revised version, we have made a number of changes to address all of their concerns. Specific answers to the reviewers are given below, and in this general reply we summarize our changes (marked in color in the revised version of our paper):

- We have modified the main text, as well as added an appendix, to better contextualize our approach with both flow matching via stochastic interpolants and score-based diffusion models. As we now explain in more detail, the approach that we propose can be used in two primary ways. The first is to *distill of the velocity field of a probability flow ODE*, which can be pre-trained using either flow matching or score-based diffusion. The second is to *train the flow map of a probability flow ODE from scratch*; in this case, the stochastic interpolant is used as data but a pre-trained velocity field is not required. These additional explanations should also address the theoretical questions raised by reviewer NFyf.
- We have extended the related work section and changed the title of our paper to better position our approach in the context of existing methods. In particular, we hope that this clarifies that our method fits within the *consistency models* paradigm, and that it provides a unifying framework that encompasses, to our knowledge, all existing consistency-based techniques in a single formulation.
- We have expanded our numerical realization section to provide more experiments that showcase the potential of our method. Note that our aim here is primarily to show the benefit of providing a unifying perspective on consistency models (new learning objectives in LMD and FMM and juxtaposing the different approaches), not necessarily to reach SOTA on generative modeling tasks involving images: we believe that the method we propose will be useful in that context too, and that it can be used at scale, but to demonstate this will require much larger computational ressources than the one we have currently access to.

We hope that, with these changes, the reviewers will now find our paper suitable for publication in TMLR.

---

### Decision · Action_Editor_c6ye · 2025-03-25

**Recommendation:** Accept as is

**Comment:**

As the reviewers acknowledge, the proposed method is novel, and the theoretical results are solid. The reviewers had several concerns regarding the theoretical contributions and presentation of the results, but the authors adequately addressed the concerns in the revised version. Although the concerns regarding the scale of the experiments remain valid, the strengths of the paper outweigh this weakness. Therefore, I recommend the paper for acceptance at TMLR.

**Audience:**

The problem considered in the paper is relevant to TMLR's audience. I believe many individuals will be interested in the findings of this paper.

**Claims And Evidence:**

The theoretical claims are solid and supported by rigorous proofs. However, the numerical experiments are done on relatively small tasks.